# Association and dissociation between the mitochondrial Far complex and Atg32 regulate mitophagy

**Aleksei Innokentev†, Kentaro Furukawa†\*, Tomoyuki Fukuda, Tetsu Saigusa, Keiichi Inoue, Shun-ichi Yamashita, Tomotake Kanki\***

Department of Cellular Physiology, Niigata University Graduate School of Medical and Dental Sciences, Niigata, Japan

**Abstract** Mitophagy plays an important role in mitochondrial homeostasis. In yeast, the phosphorylation of the mitophagy receptor Atg32 by casein kinase 2 is essential for mitophagy. This phosphorylation is counteracted by the yeast equivalent of the STRIPAK complex consisting of the PP2A-like protein phosphatase Ppg1 and Far3-7-8-9-10-11 (Far complex), but the underlying mechanism remains elusive. Here we show that two subpopulations of the Far complex reside in the mitochondria and endoplasmic reticulum, respectively, and play distinct roles; the former inhibits mitophagy via Atg32 dephosphorylation, and the latter regulates TORC2 signaling. Ppg1 and Far11 form a subcomplex, and Ppg1 activity is required for the assembling integrity of Ppg1-Far11-Far8. The Far complex preferentially interacts with phosphorylated Atg32, and this interaction is weakened by mitophagy induction. Furthermore, the artificial tethering of Far8 to Atg32 prevents mitophagy. Taken together, the Ppg1-mediated Far complex formation and its dissociation from Atg32 are crucial for mitophagy regulation.

**\*For correspondence:**
furukawa@med.niigata-u.ac.jp
(KF);
kanki@med.niigata-u.ac.jp (TK)

†These authors contributed equally to this work

**Competing interests:** The authors declare that no competing interests exist.

## Introduction

Macroautophagy (hereafter referred to as autophagy) is a catabolic process that non-selectively degrades cytoplasmic components in response to a wide range of cellular stresses, such as nutrient starvation, oxidative stress, infection, and inflammatory stimuli. Upon autophagy induction, a cup-shaped membrane vesicle called an isolation membrane or a phagophore emerges in the cytosol. The isolation membrane extends and sequesters cytoplasmic components, forming an autophago-some. The autophagosome then fuses with vacuoles in yeast or lysosomes in mammalian cells, and vacuolar/lysosomal hydrolases degrade the sequestered material (*Nakatogawa et al., 2009*). In contrast to bulk autophagy, selective autophagy targets specific cellular components, such as particular proteins, mitochondria, peroxisomes, endoplasmic reticulum (ER), nuclei, and intracellular pathogens (*Farré and Subramani, 2016*; *Gatica et al., 2018*; *Kirkin and Rogov, 2019*). Among the multiple types of selective autophagy, mitochondrial autophagy (mitophagy) has gained prominence over the last decade because quality and quantity control of mitochondria are crucial for preventing various diseases, such as cancer, diabetes, and neurodegenerative diseases (*Mizushima and Komatsu, 2011*; *Youle and Narendra, 2011*).

Recent studies have revealed the molecular mechanisms of mitophagy in yeast and mammals (*Fukuda and Kanki, 2018*). In mammals, mitophagy mediated by the mitochondrial serine/threonine protein kinase PINK1 and the E3 ubiquitin ligase Parkin and mitophagy induced by mitophagy receptors (Nix, BNIP3, FKBP8, FUNDC1, and Bcl2-L-13) have been extensively studied (*Pickles et al., 2018*). In the yeast *Saccharomyces cerevisiae*, the mitochondrial outer membrane protein Atg32 serves as a receptor essential for mitophagy (*Kanki et al., 2009*; *Okamoto et al., 2009*). When mitophagy is induced by either nitrogen starvation or cell culture in a nonfermentable medium until

the stationary phase, Ser114 and Ser119 on Atg32 are phosphorylated by casein kinase 2 (CK2) (*Kanki et al., 2013*). This phosphorylation event facilitates the interaction between Atg32 and the adaptor protein Atg11, leading to the recruitment of the core autophagy machinery, which initiates autophagosome formation around the mitochondria (*Aoki et al., 2011*). More recently, we reported that the protein phosphatase 2A (PP2A)-like protein phosphatase Ppg1 and the Far complex (consisting of Far3, Far7, Far8, Far9, Far10, and Far11) cooperatively dephosphorylate Atg32 to inhibit mitophagy (*Furukawa et al., 2018*).

The Far complex components were originally identified as factors necessary for pheromone-induced cell cycle arrest (*Horecka and Sprague, 1996*; *Kemp and Sprague, 2003*). The Far complex has also been shown to be involved in the target of rapamycin complex 2 (TORC2) signaling pathway (*Baryshnikova et al., 2010*; *Pracheil et al., 2012*) and human caspase-10-induced toxicity in yeast (*Lisa-Santamaría et al., 2012*). However, the underlying molecular mechanisms that drive these processes remain elusive. Also, the Far complex is known to require tiered assembly and localize at the ER (*Pracheil and Liu, 2013*). Still, how the ER-localized Far complex acts cooperatively with Ppg1 to affect the mitochondrial protein Atg32 remains unclear (*Furukawa and Kanki, 2018*).

Ppg1 and the Far complex form the yeast counterpart of the striatin-interacting phosphatase and kinase (STRIPAK) complex, which is widely conserved in eukaryotes (*Hwang and Pallas, 2014*). Most STRIPAK components, such as Striatin/Far8 (PP2A regulatory subunit), PP2AA/Tpd3 (PP2A scaffolding subunit), PP2Ac/Ppg1 (PP2A or PP2A-like catalytic subunit), striatin-interacting protein (STRIP)/Far11, sarcolemmal membrane-associated protein (SLMAP)/Far9-10, and suppressor of IKKε (SIKE)/Far3-7, are highly conserved in eukaryotes, whereas the other components, such as monopolar spindle one-binder protein (Mob), germinal center kinase (GCKIII), and cerebral cavernous malformation 3 (CCM3), are not fully conserved (*Goudreault et al., 2009*; *Kück et al., 2016*). The STRIPAK complex has been shown to be involved in diverse cellular processes, including development, cellular transport, signal transduction, stem cell differentiation, and cardiac functions (*Kück et al., 2019*). Despite the biochemical, structural, and physiological characterizations that have been performed on the STRIPAK complex using diverse eukaryotic model organisms, the upstream regulators and downstream effectors of the STRIPAK complex are not yet fully understood. In particular, the mechanism that regulates the enzymatic activities and cellular localization of the STRIPAK complex remains unclear.

In this study, we found that the Far complex is localized at both the mitochondria and ER and the mitochondria-localized Far complex mediates the Ppg1-dependent inhibition of mitophagy via Atg32 dephosphorylation, whereas the ER-localized Far complex plays a role in the regulation of the TORC2 signaling pathway. Remarkably, Ppg1 phosphatase activity is essential for the recruitment of the Ppg1-Far11 subcomplex to the core of the Far complex. The Far complex preferentially interacts with phosphorylated Atg32 via the 151–200 amino acid region of Atg32, and this interaction becomes weakened under mitophagy-inducing conditions. Far8 directly interacts with Atg32, and their artificial tethering prevents mitophagy. Taken together, we propose that the association and dissociation of the mitochondria-localized Far complex and Atg32 represent crucial processes that regulate mitophagy.

## Results

### Mitochondria-localized, but not ER-localized, Far complex is required for Atg32 dephosphorylation

Although Far9 and Far10 both contain tail-anchor (TA) domains required for the localization of the entire Far complex to the ER (*Pracheil and Liu, 2013*), the reason why ER-localized Far complex is essential for the dephosphorylation of the mitochondrial protein Atg32 remains unclear. Thus, we reinvestigated the cellular localization of Far proteins using yeast strains expressing Far proteins fused to either an N-terminal or a C-terminal green fluorescent protein (GFP) tag (Far3-GFP, Far7-GFP, Far8-GFP, GFP-Far9, GFP-Far10, and Far11-GFP). All GFP-fused Far proteins were confirmed to be functional as assessed by Atg32 dephosphorylation (*Figure 1—figure supplement 1A*). In addition to the ER (*Figure 1—figure supplement 1B*), we realized that a substantial portion of GFP-fused Far proteins was localized in the mitochondria (*Figure 1A*; Far8-GFP, 93%; GFP-Far9, 96%; and Far11-GFP, 94% of cells showed mitochondrial GFP signal). These findings were consistent with

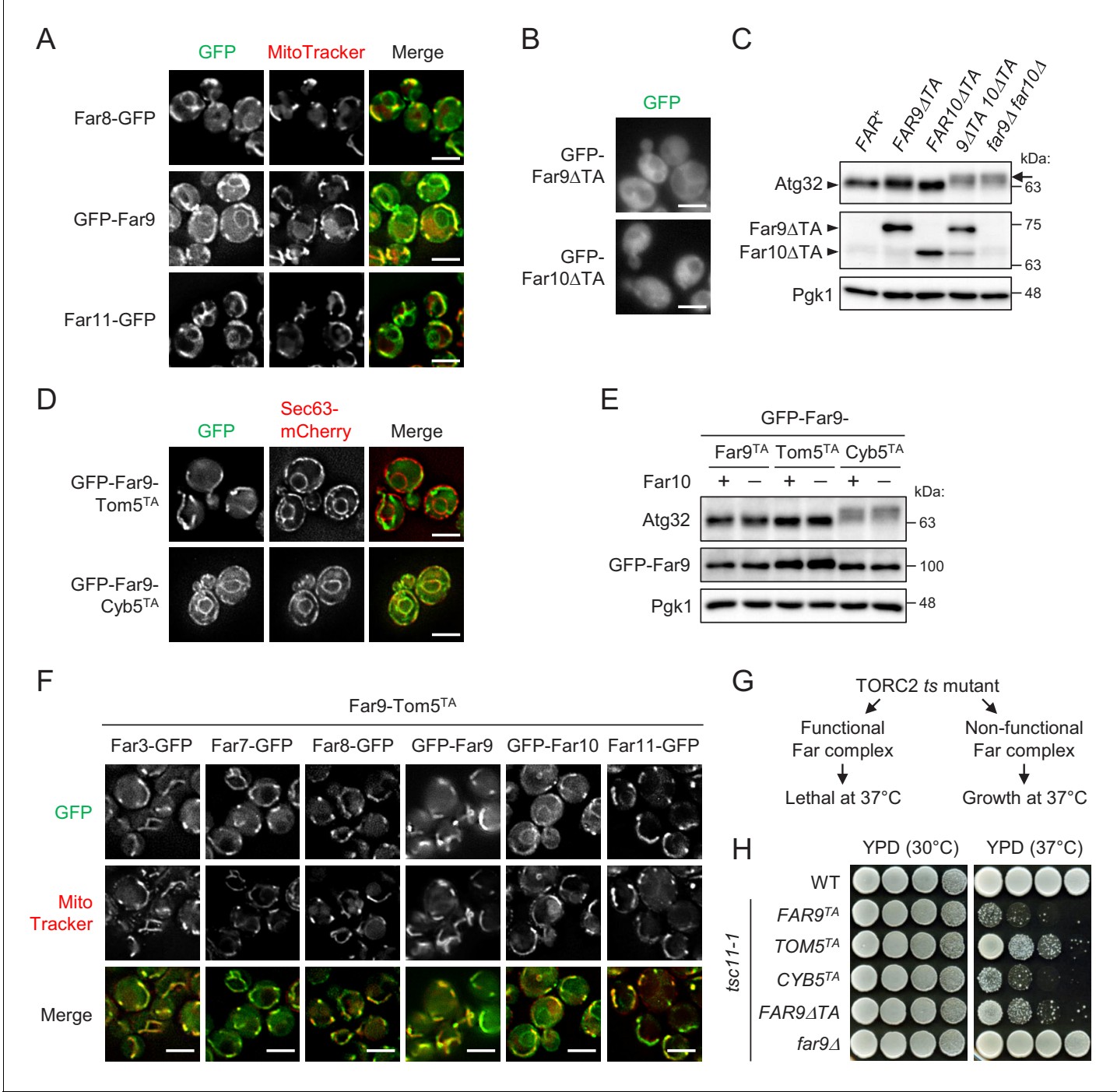

**Figure 1.** Mitochondria-localized, but not ER-localized, Far complex is required for Atg32 dephosphorylation. (A, B, D, and F) Cells expressing the indicated GFP-fused Far proteins were cultured in YPD medium until the early log growth phase and analyzed by fluorescence microscopy. Sec63-mCherry and MitoTracker Red CMXRos were used to visualize the ER and mitochondria, respectively. Representative images of at least 100 cells are shown. Scale bar, 4 μm. (C and E) The indicated cells were cultured in YPL medium until the mid-log growth phase. Atg32 status was analyzed by western blot (WB) using an anti-Atg32 antibody. Far9ΔTA and Far10ΔTA were detected using the anti-HA antibody. GFP-Far9 derivatives were detected with an anti-Far9 antibody. Pgk1 was detected as a loading control (throughout this study). For Atg32 detection, arrowhead and arrow indicate the dephosphorylated and phosphorylated Atg32, respectively. (G) Functional and non-functional Far complexes cause opposite growth phenotypes in the TORC2 ts mutant background. (H) Wild-type cells, tsc11-1 cells expressing GFP-Far9 derivatives, and tsc11-1 far9Δ cells were cultured in YPD medium until the early log growth phase. Serial dilutions of each culture were spotted on YPD agar plates and cultured at 30°C for 24 hr or 37°C for 48 hr (three independent replicates). WB experiments were independently replicated three (E) or four times (C).

*Figure 1 continued on next page*

*Figure 1 continued*

The online version of this article includes the following figure supplement(s) for figure 1:

**Figure supplement 1.** Expression, functional, and localization analyses of GFP-fused Far proteins.

previous reports showing that SLMAP and PRO45, which are the Far9/Far10 orthologues in humans and the filamentous fungus *Sordaria macrospora*, respectively, are localized at multiple cellular compartments, including the ER and mitochondria (*Byers et al., 2009*; *Nordzieke et al., 2015*).

The dual localization of Far9 at both the ER and mitochondria prompted us to investigate the physiological significance of each localization. As reported previously, GFP-Far9 lacking its TA domain (GFP-Far9ΔTA) was localized diffusely throughout the cytoplasm, and the same was true for Far10 lacking its TA domain (*Figure 1B*; GFP-Far9ΔTA, 99%; GFP-Far10ΔTA, 100% of cells showed cytoplasmic GFP signal). To examine whether the presence of the Far9 and Far10 TA domains is necessary for Atg32 dephosphorylation, we constructed single and double Far9 (Far9ΔTA) and Far10 (Far10ΔTA) mutants, whose TA domains were replaced by 3HA tags. Under growing conditions, Atg32 is dephosphorylated in wild-type cells, whereas it displays a phosphorylated form, detected as slowly migrating bands in immunoblotting experiments, in *FAR*-deficient cells (*Figure 1—figure supplement 1A*). As shown in *Figure 1C*, the Far9ΔTA and Far10ΔTA single mutant cells did not display significant alterations in the phosphorylation status of Atg32, whereas the double mutant cells showed highly phosphorylated Atg32 at the same levels as observed in cells lacking both Far9 and Far10 (*far9Δ far10Δ*). These results suggest that the membrane anchoring of at least one of Far9 or Far10 is important for the Far complex-mediated dephosphorylation of Atg32.

Next, to examine the possibility that mitochondria- and ER-localized Far complexes have different functions, we constructed two GFP-Far9 mutants, in which the TA domains were replaced by the TA domain of Tom5 and Cyb5, which localize specifically to the mitochondria and ER, respectively (*Beilharz et al., 2003*). As expected, GFP-Far9-Tom5$^{TA}$ and GFP-Far9-Cyb5$^{TA}$ showed distinct mitochondrial and ER localization, respectively (*Figure 1D*; GFP-Far9-Tom5$^{TA}$, 100% mitochondria; GFP-Far9-Cyb5$^{TA}$, 99% ER). Remarkably, Atg32 in GFP-Far9-Tom5$^{TA}$ cells appeared in the dephosphorylated form similar to Atg32 in cells expressing GFP-Far9 under growing conditions, whereas Atg32 was highly phosphorylated in GFP-Far9-Cyb5$^{TA}$ cells regardless of the presence or absence of Far10 (*Figure 1E*). We further confirmed that the expression of Far9-Tom5$^{TA}$ resulted in the almost complete recruitment of other Far proteins to the mitochondria, except for Far10 (*Figure 1F*; Far3-GFP, 99%; Far7-GFP, 98%; Far8-GFP, 99%; GFP-Far9, 96%; GFP-Far10, 0%; Far11-GFP, 99% of cells showed mitochondria-specific GFP signal). Thus, these results indicate that the mitochondria-localized, but not the ER-localized, Far complex prevents Atg32 phosphorylation.

The Far complex is genetically linked to the TORC2 signaling pathway. The deletion of the genes coding the Far complex components can suppress the growth defects caused by temperature-sensitive (*ts*) mutations of the TORC2 components (*Baryshnikova et al., 2010*; *Pracheil et al., 2012*). However, cells expressing intact Far proteins (*FAR*$^{+}$) do not suppress the *ts* phenotype. Therefore, cells expressing *ts* mutations in TORC2 components may be used to determine whether the mutated forms of the Far components are functional (*Figure 1G*). Using the *tsc11-1* (*avo3-1*) mutant (*Zinzalla et al., 2011*), we examined whether fixing the localization of Far9 to either the ER or mitochondria affects the role played by the Far complex in TORC2 signaling. As shown in *Figure 1H*, wild-type and *tsc11-1 far9Δ* cells grew normally at 37°C, whereas *tsc11-1* cells expressing GFP-Far9 or GFP-Far9-Cyb5$^{TA}$ did not grow at 37°C, indicating that GFP-Far9-Cyb5$^{TA}$ is functional. In contrast, the growth of *tsc11-1* cells expressing GFP-Far9-Tom5$^{TA}$ or GFP-Far9ΔTA at 37°C was partially restored, indicating that these proteins are not fully functional in TORC2 signaling. These findings suggest that the mitochondria- and ER-localized Far complexes differentially contribute to Atg32 dephosphorylation and TORC2 signaling, respectively.

## The mitochondria-localized Far complex is a limiting factor for the Ppg1-dependent inhibition of mitophagy via Atg32 dephosphorylation

We next investigated whether artificially fixing the localization of the Far complex to either the mitochondria or ER affects mitophagy using the Idh1-GFP processing assay (*Kanki and Klionsky, 2008*). Idh1-GFP is localized in the mitochondrial matrix and delivered into vacuoles by mitophagy.

Although Idh1-GFP is degraded by vacuolar hydrolases, the GFP moiety remains relatively stable even within the vacuole and is released as an intact protein. Therefore, mitophagy levels can be semiquantitatively monitored by measuring the levels of processed GFP using immunoblotting.

As shown in *Figure 2A*, we compared the mitophagy levels among wild-type, *FAR9-TOM5*$^{TA}$, *FAR9-CYB5*$^{TA}$, *far9Δ*, and *atg1Δ* (as a negative control) cells. We found that mitophagy and Atg32 phosphorylation in *FAR9-TOM5*$^{TA}$ cells were strongly inhibited compared with wild-type cells. In contrast, increased mitophagy was observed in *FAR9-CYB5*$^{TA}$ and *far9Δ* cells compared with wild-type cells. The effect of *FAR9-TOM5*$^{TA}$ on Atg32 dephosphorylation and mitophagy was canceled by the absence of Ppg1 (*Figure 2B*), consistent with the Far complex acting through Ppg1. Furthermore, Ppg1 overexpression did not result in additional effects on mitophagy in *FAR9-TOM5*$^{TA}$ cells (*Figure 2B*). Taken together, these results indicate that the mitochondria-localized Far complex

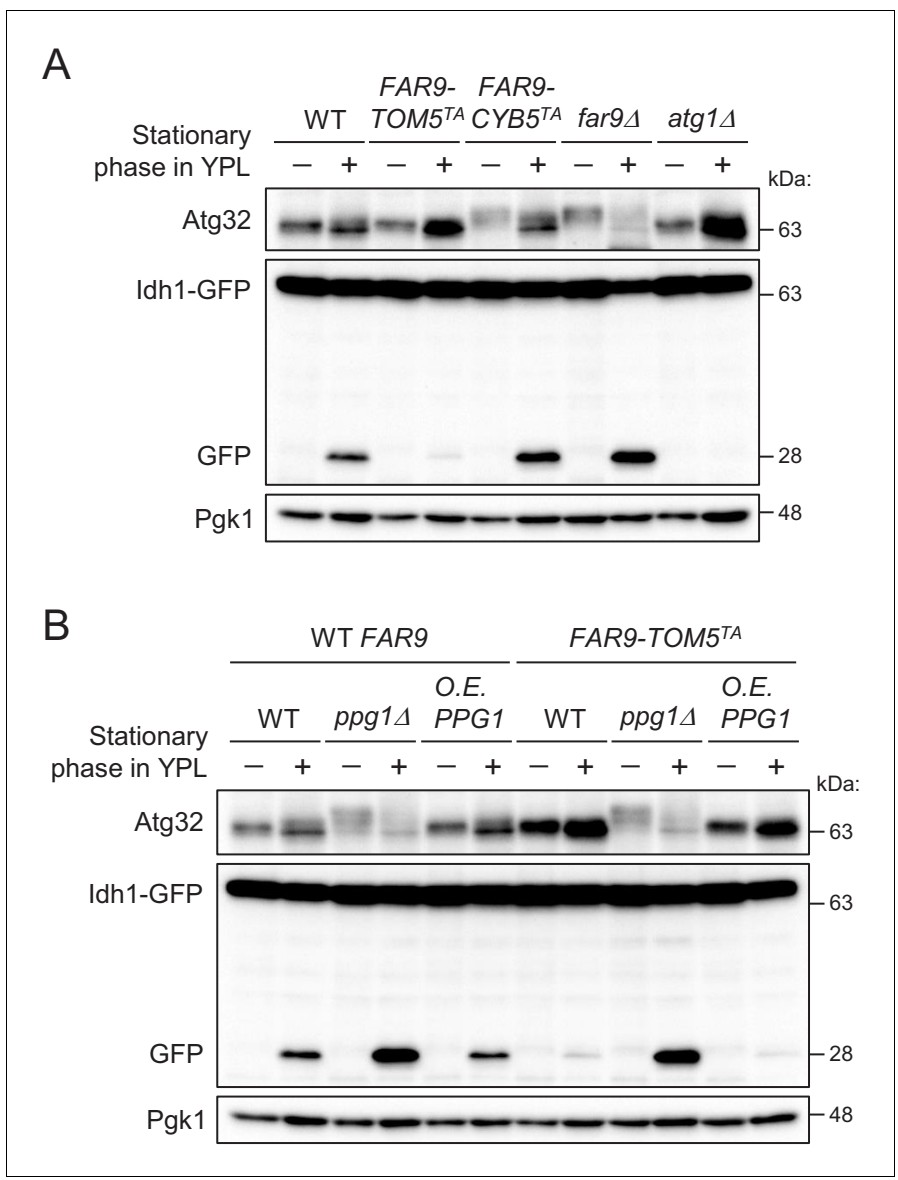

**Figure 2.** The mitochondria-localized Far complex is a limiting factor for the Ppg1-dependent inhibition of mitophagy via Atg32 dephosphorylation. (**A and B**) The indicated cells expressing Idh1-GFP were continuously cultured in YPL medium and collected at 20 hr (growing phase) and 40 hr (stationary phase). Atg32 status and Idh1-GFP processing were analyzed by Western blot (WB) with anti-Atg32 and anti-GFP antibodies, respectively. WB experiments were independently replicated three times (**A and B**).

represents a limiting factor for the Ppg1-dependent inhibition of mitophagy via Atg32 dephosphorylation.

## Ppg1, Far11, and Tpd3 form a subcomplex, but Tpd3 plays a limited role as a scaffold protein in the Ppg1-Far complex

Next, we focused on the relationship between the Far complex and Ppg1. Our previous mass spectrometric analysis identified Far8 as a Ppg1-associated protein (*Furukawa et al., 2018*). Using an immunoprecipitation assay, we confirmed that not only Far8 but also other Far proteins co-immunoprecipitated with Ppg1 (*Figure 3—figure supplement 1*). To determine which Far protein is the primary binding partner for Ppg1, we constructed *ppg1Δ* strains lacking one or all of the *FAR* genes (*far3Δ far7Δ far8Δ far9Δ far10Δ far11Δ*) that co-express FLAG-His6-Ppg1 and 3HA- or GFP-fused Far proteins. Using these strains, immunoprecipitation assays with an anti-FLAG affinity gel was performed (*Figure 3A–F*). We expected that the primary binding partner of Ppg1 would interact with Ppg1 even in the absence of the other Far proteins. As shown in *Figure 3F*, only Far11-3HA among the six Far proteins co-immunoprecipitated with FLAG-His6-Ppg1 in all of the *FAR* deletion backgrounds. These results suggest that Ppg1 primarily binds to Far11.

The heterotrimeric PP2A complex is composed of catalytic (C), scaffold (A), and variable regulatory subunits (B, B′, B″, and B‴). The STRIPAK complex contains the C, A, and B‴ subunits and the A subunit is thought to be essential for connecting the C subunit with the B‴ subunit (*Goudreault et al., 2009*). Although our previous mass spectrometric analysis identified Tpd3 (A subunit) as a Ppg1-binding protein, Tpd3 was found to be dispensable for Atg32 dephosphorylation (*Furukawa et al., 2018*). A recent report regarding the STRIPAK complex in *Aspergillus nidulans* showed that SipE (Ppg1 orthologue), SipC (Far11 orthologue), and SipF (Tpd3 orthologue) form a heterotrimeric subcomplex (*Elramli et al., 2019*). Therefore, we decided to investigate the relationship among Ppg1, Far11, and Tpd3 in greater depth. Using immunoprecipitation assays with an anti-FLAG affinity gel, we first verified that Ppg1 (FLAG-His6-Ppg1) indeed interacts with Tpd3 (Tpd3-3HA) as well as Far11 (*Figure 3G*). The interactions between Ppg1 and Tpd3 and between Ppg1 and Far11 were not disrupted in the *far11Δ* and *tpd3Δ* backgrounds, respectively (*Figure 3G*). In contrast, the interaction between Far11 and Tpd3 was completely disrupted in the *ppg1Δ* background (*Figure 3H*). These results suggest that Ppg1, Far11, and Tpd3 form a subcomplex, and that Ppg1 is a central factor in this subcomplex.

Next, we performed immunoprecipitation assays using an anti-Far8 antibody to investigate whether Tpd3 is included in the Ppg1-Far complex. As shown in *Figure 3I*, Tpd3 co-immunoprecipitated with Far8 much less efficiently than Far11. Together with the previously reported dispensable role of Tpd3 in Atg32 dephosphorylation, this result suggests that Tpd3 is not a scaffold protein in the Ppg1-Far complex and that the majority of Tpd3 interacts with Ppg1 without Far complex or PP2A (Pph21 and Pph22) (*van Zyl et al., 1992*) for the mitophagy-independent process. During these analyses, we found that the co-immunoprecipitation of Far11 and Tpd3 with Far8 depends on Ppg1 (*Figure 3I*), suggesting that Ppg1 is required for the assembly of the Far complex (*Figure 3J*).

## Ppg1 phosphatase activity is required for the assembling integrity of Ppg1-Far11-Far8

The above data suggest that Ppg1 is required for the interaction between Far8 and Far11. Next, we examined whether Ppg1 is also required for the interactions between Far8 and the other Far proteins. We constructed yeast strains co-expressing 3HA- or GFP-fused Far proteins (Far3-3HA, Far7-3HA, and GFP-Far9) in *PPG1+* and *ppg1Δ* backgrounds to explore the interactions. Immunoprecipitation assays with an anti-Far8 antibody showed that the interactions between Far8 and Far3, Far7, Far9, and Far11 all occurred in the *PPG1+* background, as expected; however, only the Far8-Far11 interaction was disrupted in *ppg1Δ* cells (*Figure 4A*). Moreover, we found that only Ppg1 among the various PP2A family proteins (Pph21, Pph22, Pph3, Sit4, and Ppg1) is required for the interaction between Far8 and Far11 (*Figure 4B*).

Next, we examined whether Ppg1 affects the localization of the Far proteins. In cells lacking Ppg1, Far11-GFP, but not the other Far components, was diffused throughout the entire cytoplasm (*Figure 4C*; Far3-GFP, 1%; Far7-GFP, 1%; Far8-GFP, 1%; GFP-Far9, 2%; GFP-Far10, 0%; Far11-GFP, 100% of cells showed cytoplasmic GFP signal). The dislocalization of Far11 in the absence of Ppg1

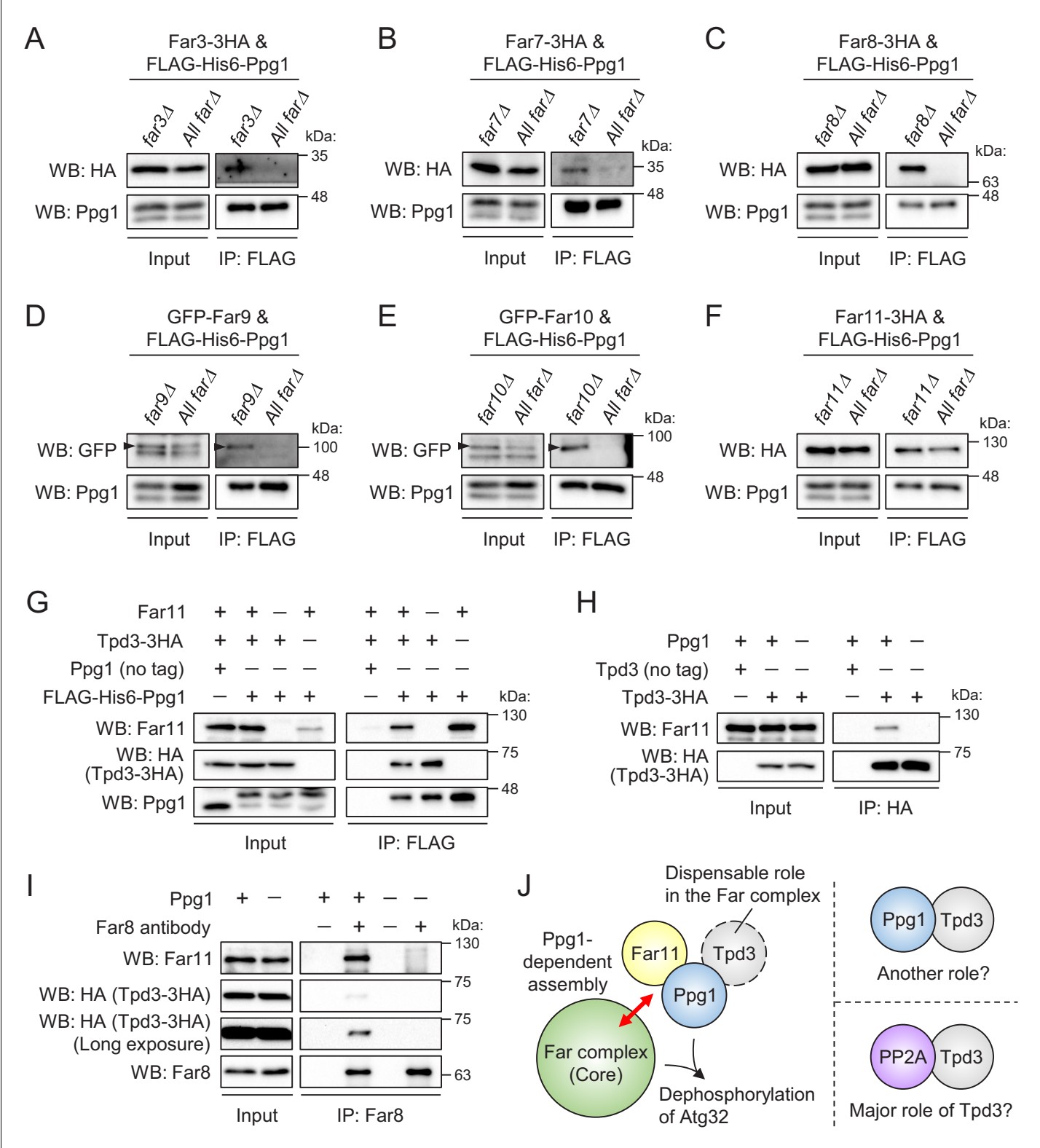

**Figure 3.** Ppg1, Far11, and Tpd3 form a subcomplex, but Tpd3 plays a limited role as a scaffold protein in the Ppg1-Far complex. The indicated cells co-expressing FLAG-His6-Ppg1 and Far3-3HA (**A**), Far7-3HA (**B**), Far8-3HA (**C**), GFP-Far9 (**D**), GFP-Far10 (**E**), or Far11-3HA (**F**) were cultured in SMD-Trp-Ura (A–C and F) or SMD-Ura (D and E) medium until the early log growth phase. FLAG-His6-Ppg1 was precipitated from cell lysates using an anti-FLAG M2 affinity gel. Cell lysates (Input) and precipitates (IP: FLAG) were analyzed by western blot (WB) with anti-HA, anti-GFP, and anti-Ppg1 antibodies. (**G**) *ppg1Δ TPD3-3HA*, *ppg1Δ far11Δ TPD3-3HA*, and *ppg1Δ tpd3Δ* cells expressing Ppg1 (no tag) or FLAG-His6-Ppg1 were cultured in SMD-Ura medium

*Figure 3 continued on next page*

*Figure 3 continued*

until the early log growth phase. Cell lysates (Input) and anti-FLAG immunoprecipitates (IP: FLAG) were analyzed by WB with anti-Far11, anti-HA, and anti-Ppg1 antibodies. (H) Wild-type, *TPD3-3HA*, and *ppg1Δ TPD3-3HA* cells were cultured in YPD medium until the early log growth phase. Cell lysates (Input) and anti-HA immunoprecipitates (IP: HA) were analyzed by WB with anti-Far11 and anti-HA antibodies. (I) *TPD3-3HA* and *ppg1Δ TPD3-3HA* cells were cultured in YPD medium until the early log growth phase. Cell lysates (Input) and anti-Far8 immunoprecipitates (IP: Far8) were analyzed by WB with anti-Far11, anti-HA, and anti-Far8 antibodies. (J) Ppg1, Far11, and Tpd3 form a subcomplex, and this subcomplex binds to the core of the Far complex in a Ppg1-dependent manner. Tpd3 is dispensable for the Ppg1-Far complex, although Ppg1 and Tpd3 may play a Far complex-independent role. The major role of Tpd3 might be a scaffold protein of PP2A (Pph21 and Pph22) rather than that of Ppg1. WB experiments were independently replicated three times (A–I).

The online version of this article includes the following figure supplement(s) for figure 3:

**Figure supplement 1.** Ppg1 interacts with the Far3-7-8-9-10-11 proteins.

was also observed in cells expressing the mitochondria-fixed Far9-Tom5$^{TA}$ (*Figure 4D*; *PPG1$^+$*, 1%; *ppg1Δ*, 97% of cells showed cytoplasmic GFP signal). We found that the interaction of Ppg1 with Far8 is abolished in the absence of Far11 (*Figure 4E*). As the interaction of Far11 with Far8 requires Ppg1 (*Figure 4A and B*), it is likely that the subcomplex formation between Ppg1 and Far11 is a prerequisite for them to bind to the core of the Far complex.

We further investigated whether the catalytic activity of Ppg1 affects the assembly of the Far complex. Co-immunoprecipitation of Far11 with Far8 was strongly impaired in cells expressing a catalytically inactive Ppg1-H111N mutant (*Figure 4F*). Ppg1 activity was also required, but not essential, for the interaction between Ppg1 and Far8/Far11 (*Figure 4G*). In summary, the phosphatase activity of Ppg1 is required for the assembling integrity of Ppg1-Far11-Far8 (core Far proteins).

## Interaction between the Far complex and phosphorylated Atg32 is impaired under mitophagy-inducing conditions

In wild-type cells, Atg32 phosphorylation is prevented by the Ppg1-Far complex under growing conditions, but Atg32 is rapidly phosphorylated by CK2 upon mitophagy induction. As CK2 is a constitutively active kinase (*Litchfield, 2003*), we hypothesized that a mechanism exists to suppress the function of the Ppg1-Far complex under mitophagy-inducing conditions (*Furukawa et al., 2018*). To investigate this possibility, we performed two immunoprecipitation assays to analyze the interactions between Ppg1 and Far8/Far11 and between Far8 and Far11. As shown in *Figure 5A and B*, respectively, these interactions were not altered either before or after mitophagy induction by starvation. Also, the localization of the Far proteins, except for Far10, to either the ER or mitochondria did not change after starvation (*Figure 5—figure supplements 1* and *2*). Thus, the promotion of Atg32 phosphorylation upon mitophagy induction is not attributed to the disintegration or translocation of the Far complex.

We next investigated whether the Far complex interacts with Atg32 and, if so, how this interaction is regulated. We expressed N-terminally 3HA-tagged Atg32 (3HA-Atg32) in *atg32Δ PPG1$^+$* and *atg32Δ ppg1Δ* backgrounds and performed immunoprecipitation assays using anti-Far8 or anti-HA antibody. As shown in *Figure 5C*, reciprocal immunoprecipitation demonstrated the interaction between Atg32 and Far8. Importantly, this co-immunoprecipitation was dramatically enhanced in *ppg1Δ* cells, in which Atg32 is constitutively phosphorylated, suggesting that the Far complex preferentially interacts with a phosphorylated form of Atg32. To confirm this finding, we expressed a non-phosphorylatable form of Atg32 (2SA: S114A/S119A mutation; *Figure 5D*) in *ppg1Δ* cells and performed an immunoprecipitation assay. As expected, the co-immunoprecipitation of Far8 with Atg32-2SA was remarkably decreased compared to wild-type Atg32 (*Figure 5E*). In contrast, a phospho-mimic mutation (S114D/S119D) did not enhance the interaction, and the interaction was much less than that between truly phosphorylated Atg32 and Far8 (*Figure 5—figure supplement 3A*). This result is consistent with our previous report that the SD mutant fails to mimic the phosphorylated form of Atg32 (*Aoki et al., 2011*). Based on these data, we used the *ppg1Δ* background in subsequent experiments to efficiently detect the interaction between Atg32 and the Far complex.

Next, we investigated which Far proteins are required for the interaction between the Far complex and Atg32. We performed immunoprecipitation assays with an anti-HA antibody using *atg32Δ ppg1Δ* strains expressing 3HA-Atg32 and lacking different *FAR* genes. As shown in *Figure 5F*, the interaction between Atg32 and Far8 was completely disrupted in *far3Δ*, *far7Δ*, and *far9Δ* cells, as

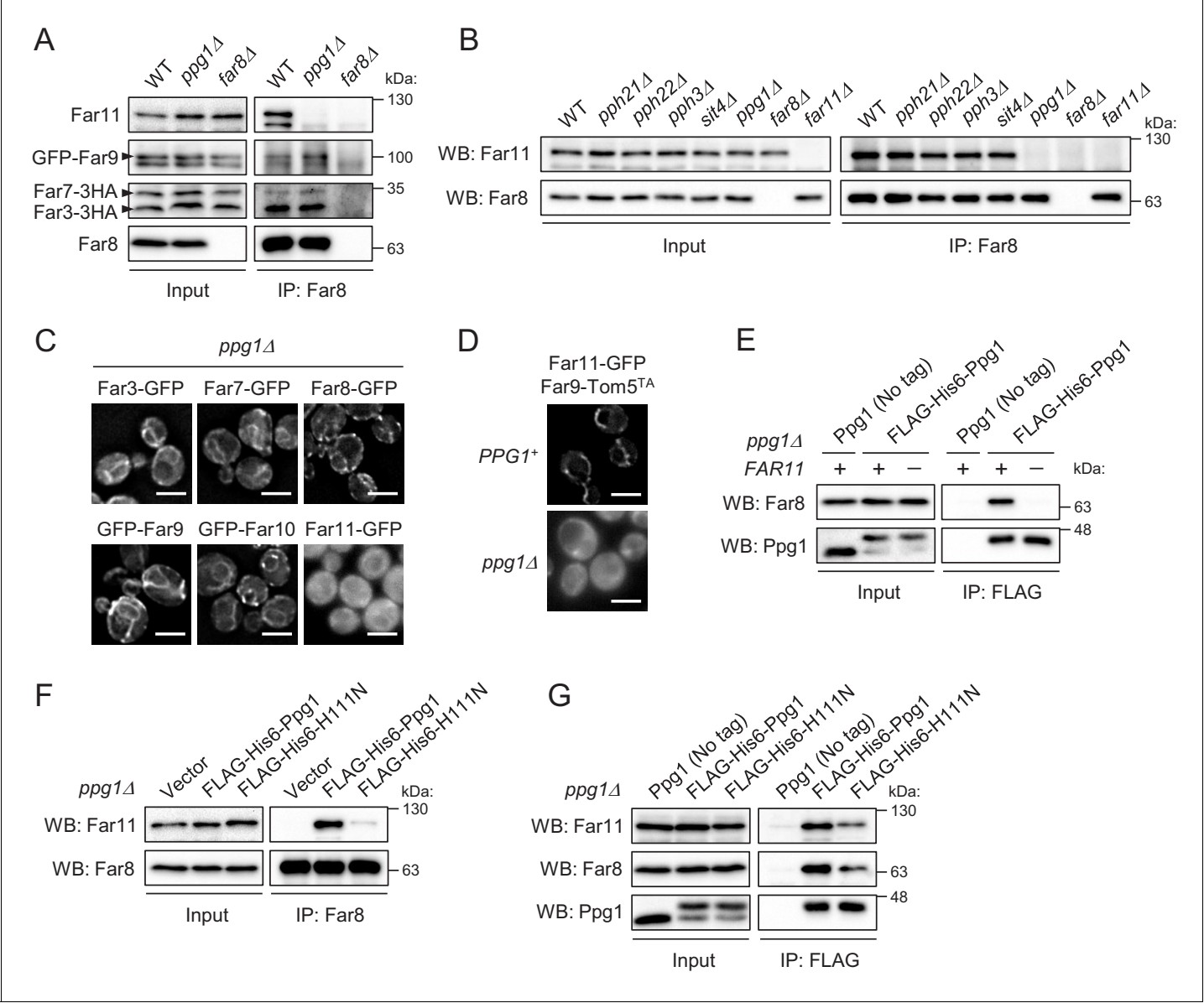

**Figure 4.** Ppg1 phosphatase activity is required for the assembling integrity of Ppg1-Far11-Far8. (**A**) Wild-type, *ppg1Δ*, and *far8Δ* (negative control) cells expressing HA- or GFP-tagged Far proteins were cultured in YPD medium until the early log growth phase. Cell lysates (Input) and anti-Far8 immunoprecipitates (IP: Far8) were analyzed by western blot (WB) with anti-HA, anti-Far8, anti-Far9, and anti-Far11 antibodies. (**B**) The indicated cells were cultured in YPL medium until the mid-log growth phase. Atg32 status was analyzed by WB with an anti-Atg32 antibody. (**C**) *ppg1Δ* cells expressing the indicated GFP-fused Far proteins were cultured in YPD medium until the early log growth phase and analyzed by fluorescence microscopy. Representative images of at least 100 cells are shown. Scale bar, 4 μm. (**D**) *PPG1*⁺ and *ppg1Δ* cells expressing Far9-Tom5^TA and Far11-GFP were cultured in YPD medium until the early log growth phase and analyzed by fluorescence microscopy. Representative images of at least 100 cells are shown. (**E and G**) *ppg1Δ* or *ppg1Δ far11Δ* cells expressing the indicated Ppg1 derivatives were cultured in SMD-Ura medium until the early log growth phase. FLAG-His6-Ppg1 was precipitated from cell lysates using an anti-FLAG M2 affinity gel. Cell lysates (Input) and anti-FLAG immunoprecipitates (IP: FLAG) were analyzed by WB with anti-Far8, anti-Far11, and anti-Ppg1 antibodies. (**F**) *ppg1Δ* cells expressing the indicated Ppg1 derivatives (empty vector as a negative control) were cultured in SMD-Ura medium until the early log growth phase. Cell lysates (Input) and anti-Far8 immunoprecipitates (IP: Far8) were analyzed by WB with anti-Far11 and anti-Far8 antibodies. WB experiments were independently replicated three (A, B, E, and F) or five times (**G**).

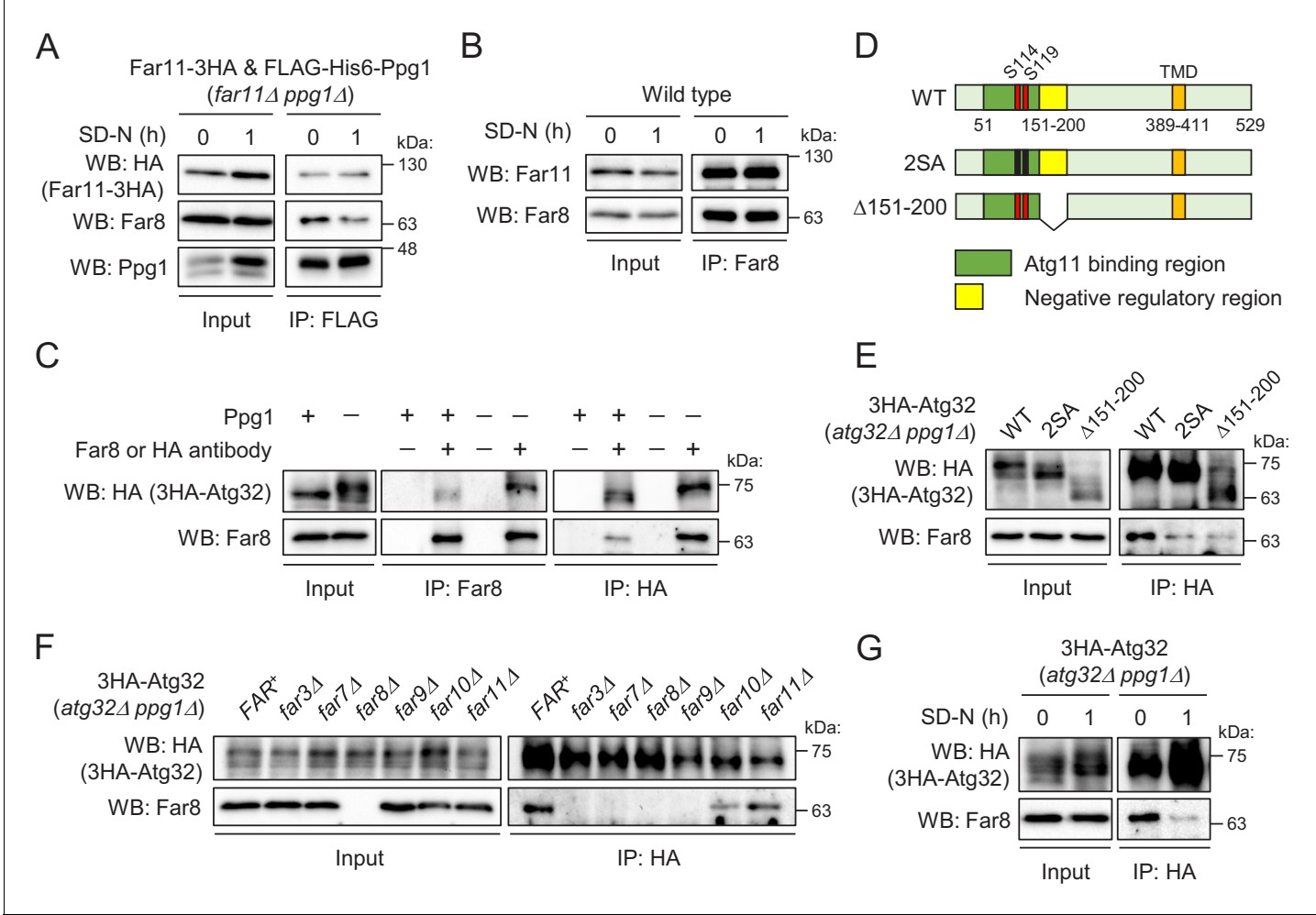

**Figure 5.** Interaction between the Far complex and phosphorylated Atg32 is impaired under mitophagy-inducing conditions. (**A**) *far11Δ ppg1Δ* cells expressing Far11-3HA and FLAG-His6-Ppg1 were cultured in SMD-Trp-Ura medium until the early log growth phase, and the cells were then shifted to SD-N for 1 hr. Cell lysates (Input) and anti-FLAG immunoprecipitates (IP: FLAG) were analyzed by western blot (WB) with anti-HA, anti-Far8, and anti-Ppg1 antibodies. (**B**) Wild-type cells were cultured in YPD medium until the early log growth phase, and the cells were then shifted to SD-N medium for 1 hr. Cell lysates (Input) and anti-Far8 immunoprecipitates (IP: Far8) were analyzed by WB with anti-Far11 and anti-Far8 antibodies. (**C**) *atg32Δ* and *atg32Δ ppg1Δ* cells expressing 3HA-Atg32 were cultured in SMD-Ura medium until the early log growth phase. Cell lysates (Input), anti-Far8 immunoprecipitates (IP: Far8), and anti-HA immunoprecipitates (IP: HA) were analyzed by WB with anti-HA and anti-Far8 antibodies. (**D**) Schematic diagram of Atg32 and its derivatives. TMD, transmembrane domain; S114 and S119, serine residues phosphorylated by CK2; 2SA, S114A/S119A mutant; Δ151–200, Atg32 lacking the 151–200 amino acid region. (**E–G**) The indicated mutant cells expressing 3HA-Atg32 or its derivatives were cultured in SMD-Ura medium until the early log growth phase, and the cells were then shifted to SD-N medium for 1 hr (**G**). Cell lysates (Input) and anti-HA immunoprecipitates (IP: HA) were analyzed by WB with anti-HA and anti-Far8 antibodies. WB experiments were independently replicated three (**A**, **B**, **F**, and **G**) or four times (**C** and **E**).

The online version of this article includes the following figure supplement(s) for figure 5:

**Figure supplement 1.** Starvation does not affect the localization of the Far proteins.
**Figure supplement 2.** Starvation does not affect the localization of the Far proteins.
**Figure supplement 3.** Analysis of the interaction between Atg32 and the Far complex.

well as in *far8Δ* cells, which represented the negative control. This result suggests that the assembly of the core complex consisting of Far3, Far7, Far8, and Far9 is necessary for the interaction of the Far complex with Atg32.

We previously showed that the deletion of the 151–200 amino acid region of Atg32 resulted in the same effects as the absence of Ppg1 or Far complex components, suggesting that the Ppg1-Far complex dephosphorylates Atg32 using this region as a scaffold (*Furukawa et al., 2018*). Indeed,

the interaction between Far8 and Atg32 was impaired when Atg32 lacked the 151–200 amino acid region (Δ151–200; *Figure 5D and E*). Finally, we determined whether the interaction between Atg32 and Far8 was affected by mitophagy induction and found that this interaction was markedly decreased upon starvation (*Figure 5G*) or rapamycin treatment (*Figure 5—figure supplement 3B*). Taken together, our data suggest that the Ppg1-Far complex binds to Atg32 to dephosphorylate Atg32 and that the dissociation of the Far complex from Atg32 upon mitophagy induction is a fundamental mechanism that promotes Atg32 phosphorylation and the further progression of mitophagy.

## Far8 directly interacts with Atg32, and their artificial tethering prevents mitophagy

Although the above data showed the in vivo interaction between Atg32 and Far8, one of the Far complex components, it was unclear whether the interaction was direct or indirect. Therefore, we next attempted to prove their direct interaction. To this end, we performed a GST pull-down assay using purified recombinant proteins, His6-Far8 (*Figure 6A*) and GST-Atg32 (three N-terminally truncated derivatives, N250, N200, and N150), produced in *Escherichia coli*. As shown in *Figure 6B*, GST-Atg32(N250), but not GST only or GST-Atg32(N200/N150), pulled down His6-Far8. This result demonstrates that Far8 directly interacts with Atg32 and that at least the 201–250 region of Atg32 is required for the interaction. Together with our in vivo data that Atg32Δ151–200 does not efficiently interact with Far8 (*Figure 5E*), we concluded that Far8 interacts with the 151–250 region of Atg32.

The finding of the direct interaction between Far8 and Atg32 prompted us to examine whether artificial tethering of these proteins would inhibit Atg32 phosphorylation and mitophagy. To this end, we expressed a chimeric protein consisting of Far8 and Atg32 (Far8-Atg32) in $PPG1^+$ $FAR8^+$, their single or double deletion mutant. The Atg32-Far8 chimeric protein was detected as an unphosphorylated form irrespective of endogenous Far8 expression (*Figure 6C*). The dephosphorylation of Atg32-Far8 requires Ppg1 (*Figure 6C*) as well as other Far complex components, except for Far10 (*Figure 6D*). Moreover, Far11-GFP was diffused throughout the entire cytoplasm in *far8Δ* cells, while it was localized to mitochondria in the *far8Δ* cells expressing Far8-Atg32, but not wild-type Atg32 (*Figure 6E*; Far11-GFP localization in $FAR8^+$, 99% mitochondria/ER; *far8Δ*, 93% cytoplasm; *far8Δ/ ATG32*, 95% cytoplasm; *far8Δ/FAR8-ATG32*, 84% mitochondria). Thus, these results indicate that the Atg32-Far8 chimeric protein normally serves as a Far complex component and counteracts with the phosphorylation of the Atg32 part. We then addressed the impact of the artificial tethering of Atg32-Far8 on mitophagy. During the stationary phase, in wild-type cells, Atg32 was phosphorylated and mitophagy was induced, whereas in cells expressing Far8-Atg32, Far8-Atg32 remained unphosphorylated and mitophagy induction was severely compromised (*Figure 6F*). The elimination of Ppg1 restored the phosphorylation of Far8-Atg32 and mitophagy (*Figure 6F*), confirming that Far8-Atg32 is functional as a mitophagy receptor unless dephosphorylated by Ppg1. Taken together, these results indicate that association between the Far complex and Atg32 strictly inhibits Atg32 phosphorylation and mitophagy and that the dissociation of the Far complex from Atg32 is a crucial step for Atg32 phosphorylation and mitophagy.

## Discussion

Although our previous study showed that Ppg1 and the Far complex played essential roles in the prevention of CK2-mediated Atg32 phosphorylation (*Furukawa et al., 2018*), the underlying mechanism and regulation of this process remained largely unclear. In the present study, we first demonstrated that the Far complex is localized at both the mitochondria and ER and that these complexes play distinct roles; the mitochondria-localized Far complex inhibits mitophagy through the Ppg1-dependent dephosphorylation of Atg32, whereas the ER-localized Far complex negatively regulates the TORC2 signaling pathway. Second, we demonstrated that Ppg1 phosphatase activity is required for the assembly of the entire Far complex. Finally, we found that the association and dissociation between the Far complex and Atg32 are crucial determinants for mitophagy regulation. Based on these findings, we propose a model for the Ppg1-Far complex-mediated phosphoregulatory mechanism of Atg32, as shown in *Figure 7*. Our findings regarding the localization, assembly, and substrate interactions of the Far complex will serve as a model system of the study of STRIPAK complexes in the other organisms.

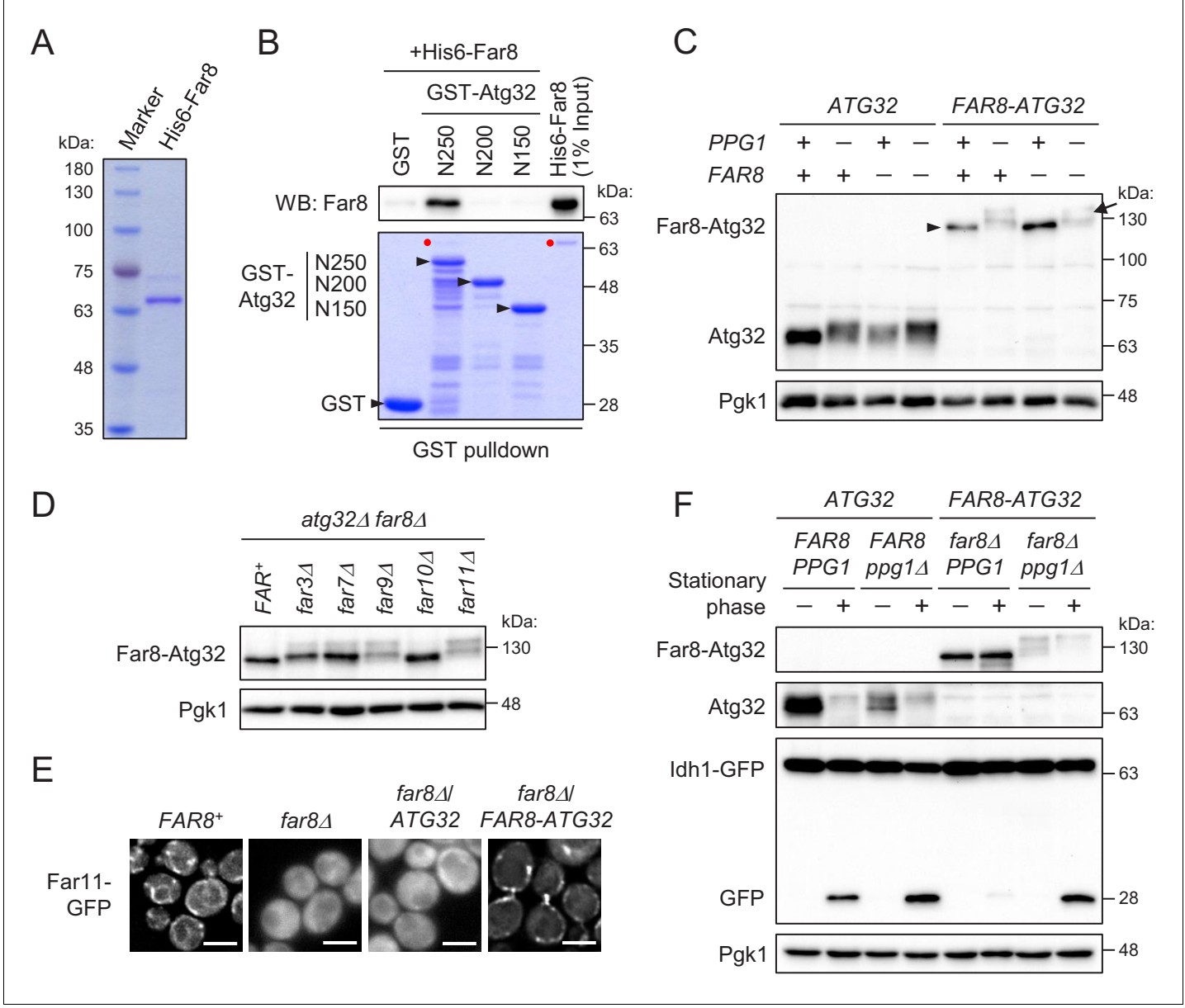

**Figure 6.** Far8 directly interacts with Atg32, and their artificial tethering prevents mitophagy. (**A**) Purification of recombinant His6-Far8 protein produced in *E. coli*. (**B**) GST pull-down analysis of the interaction between Far8 and Atg32 derivatives. GST pull-down samples were loaded on an SDS-PAGE gel followed by CBB staining or western blot (WB) with an Far8 antibody. Purified His6-Far8 protein was loaded as an input sample. Pull-down experiments were replicated three times. Red dots indicate His6-Far8. (**C and D**) The indicated cells were cultured in SMD-Ura medium until the mid-log growth phase. Atg32/Far8-Atg32 status was analyzed by WB with an anti-Atg32 antibody. For Far8-Atg32 detection, arrowhead and arrow indicate the dephosphorylated and phosphorylated Far8-Atg32, respectively (**C**). (**E**) The indicated cells expressing Far11-GFP were cultured in YPD or SMD-Ura medium until the early log growth phase and analyzed by fluorescence microscopy. Representative images of at least 100 cells are shown. Scale bar, 4 μm. (**F**) The indicated cells expressing Idh1-GFP were continuously cultured in SML-Ura medium and collected at 24 hr (growing phase) and 48 hr (stationary phase). Atg32/Far8-Atg32 status and Idh1-GFP processing were analyzed by WB with anti-Atg32 and anti-GFP antibodies, respectively. WB experiments were independently replicated three (**C and F**) or four times (**D**).

In addition to the ER localization of the Far complex (*Pracheil and Liu, 2013*), we detected its mitochondria localization and further demonstrated that the mitochondria- and ER-localized Far complexes play different roles in Atg32 dephosphorylation and TORC2 signaling, respectively (*Figure 1*). To our knowledge, this is the first report to distinguish localization-dependent roles of the Far/STRIPAK complex. Moreover, we succeeded in altering the cellular localization of the Far

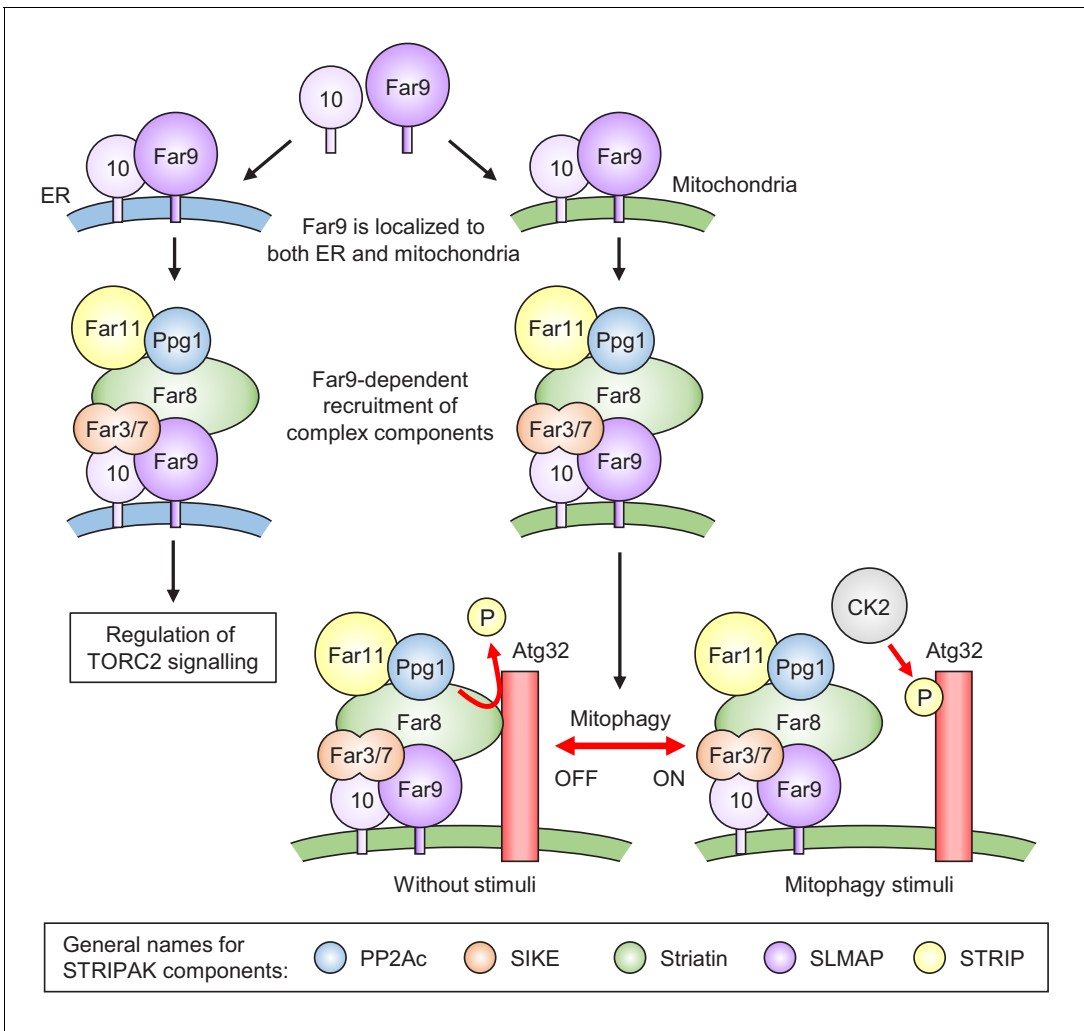

**Figure 7.** Model for the phosphoregulatory mechanism of Atg32. Far9 is localized to both the ER and mitochondria, where the other Far complex components are assembled dependently on Far9. According to this localization pattern, the Far complex plays distinct roles in the regulation of TORC2 signaling at the ER and the regulation of mitophagy at the mitochondria. Without stimuli, the mitochondria-localized Far complex mediates the Ppg1-dependent Atg32 dephosphorylation via interaction with Atg32. Upon mitophagy stimuli, the interaction between Atg32 and the Far complex is impaired, allowing Atg32 to be phosphorylated by CK2. The general names for the common STRIPAK complex components are shown in the box.

complex by replacing the TA domain of Far9. Although Far10 is a paralogue of Far9 and contains a TA domain, the mitochondria-localized Far9 recruited the other Far proteins to the mitochondria, except for Far10, indicating that Far9 is dominant to Far10 for the assembly of the other Far proteins. This hierarchy between Far9 and Far10 is consistent with their degrees of contribution to Atg32 dephosphorylation and TORC2 signaling, where Far9 plays a crucial role, but Far10 does not (*Furukawa et al., 2018*; *Pracheil et al., 2012*).

We found that artificially fixing the mitochondria localization of Far9 causes the strong inhibition of mitophagy in a Ppg1-dependent manner (*Figure 2*). This result raises the question of what regulates the balance between mitochondria- and ER-localized Far complexes in wild-type cells. Evaluation of the percentages of each localization pattern under different conditions and detailed insights into the underlying mechanisms associated with each localization are necessary to better understand this process.

PP2AA (PP2A scaffolding subunit) is generally included as a component of the STRIPAK complex. However, our results indicate that only a small portion of Tpd3, a PP2AA protein, is included in the Ppg1-Far complex (*Figure 3I*) and that Tpd3 is dispensable for Atg32 dephosphorylation (*Furukawa et al., 2018*). In contrast, Tpd3 and the Far complex components are involved in the

regulation of TORC2 signaling (*Pracheil et al., 2012*). Thus, Tpd3 is likely to be an essential component of the subpopulation of the ER-localized Ppg1-Far complex, whereas it may act as an accessory component in the mitochondria-localized population. Although Pph21 and Pph22 (PP2A catalytic subunits) have been proposed to be involved in the regulation of TORC2 signaling as components of a protein complex that includes Far proteins (*Pracheil et al., 2012*), their mutants do not disrupt the Far8-Far11 interaction as observed for the *ppg1Δ* mutant (*Figure 4B*). Also, Pph21 and Pph22 are dispensable for Atg32 dephosphorylation (*Furukawa et al., 2018*). A minor fraction of the protein complex that includes the Far proteins, Tpd3, and Pph21/Pph22 may contribute to TORC2 signaling. Alternatively, Pph21/Pph22 may regulate TORC2 signaling independently of the Far complex. Further analysis is necessary to clarify the differential actions of different subpopulations of the Far complex.

*Pracheil and Liu, 2013* proposed a tiered assembly model for the Far complex, in which a subcomplex containing Far3, Far7, and Far8 is recruited to Far9-Far10 followed by the loading of Far11 to the core (Far3-Far7-Far8-Far9-Far10). Our results indicate that Ppg1 is required for the final step of this assembly model. Moreover, the necessity of Ppg1 phosphatase activity for complex assembly implies that Far11 and/or the other Far proteins are regulated by (de)phosphorylation. In the case of Sit4, another PP2A-like protein phosphatase, its binding proteins (Sap155, Sap185, and Sap190) were previously shown to be phosphorylated in the absence of Sit4 (*Luke et al., 1996*). Therefore, further studies are necessary to investigate whether Far proteins are phosphorylated in *ppg1Δ* cells or under mitophagy-inducing conditions and whether and how such modification(s) affect the function, assembly, and localization of the Far complex.

This study identified the following five points regarding the interaction between the Far complex and Atg32 (*Figures 5* and *6*). (1) The Far complex preferentially binds to phosphorylated Atg32 rather than non-phosphorylated Atg32. (2) The 151–250 amino acid region of Atg32 acts as a scaffold for the interaction. (3) The interaction is weakened under mitophagy-inducing conditions. (4) Far8 directly interacts with Atg32 in vitro, but the core proteins of the Far complex (Far3-Far7-Far9) are also required for the interaction in yeast. (5) The forced interaction between Atg32 and the Far complex by their artificial tethering causes the inhibition of Atg32 phosphorylation and mitophagy. The first and fourth points may explain the reason for the weak in vitro interaction between Atg32 and Far8 (*Figure 6B*), which was examined using non-phosphorylated Atg32 (produced in *E. coli*) and Far8 only among six Far proteins. In addition, mitochondrial localization of Atg32 might be important for the stable interaction with Far8. Also, our results indicate that the interactions between Ppg1 and Far8/Far11 and between Far8 and Far11 are not affected by mitophagy induction. The modification(s) of any of the core proteins (Far3-Far7-Far8-Far9) may serve as a key determinant for the association/dissociation between the Far complex and Atg32. Additional experiments that examine modifications of Far proteins are necessary to solve these issues.

Recent studies of the STRIPAK complexes in mammals (*Tang et al., 2019*) and *A. nidulans* (*Elramli et al., 2019*) proposed a common model, in which two subcomplex arms converge on Striatin (Far8 orthologue). In both cases, one arm contains STRIP1/SipC (Far11 orthologue) and the other contains SIKE1-SLMAP/SipB-SipD (Far3/Far7 and Far9/Far10 orthologues, respectively). Moreover, protein kinases and Mob proteins generally included in the STRIPAK complex have not been identified yet in some organisms, including yeast. These differences in constitution among species may reflect divergent functions and regulations for the STRIPAK complexes found in each organism. STRIPAK complexes have been shown to play distinct roles in different organisms, likely due to functional repurposing (*Frost et al., 2012*).

This study revealed several previously unsolved mechanistic issues regarding the Ppg1-Far complex and provided significant insights into the research field concerned with the STRIPAK complex. However, the following issues remained unclear. First, how does Ppg1 phosphatase activity facilitate the Far complex formation? Second, what signals disrupt the interaction between the Far complex and Atg32 upon mitophagy induction? Third, are the regulations that govern the Far complex assembly and the interaction between Atg32 and the Ppg1-Far complex mediated by phosphorylation or other post-translational modifications? Future studies examining the regulatory mechanisms associated with the Ppg1-Far complex in yeast will elucidate these issues and provide detailed insights into not only how mitophagy-inducing signal is sensed and transmitted to Atg32 but also the evolutionarily conserved mechanism associated with the assembly, localization, and activation of the STRIPAK complex.

# Materials and methods

**Key resources table**

| Reagent type (species) or resource | Designation | Source or reference | Identifiers | Additional information |
|---|---|---|---|---|
| Strain, strain background (*S. cerevisiae*) | SEY6210 | DOI: 10.1128/mcb.8.11.4936 | | *MATα leu2-3,112 ura3-52 his3-Δ200 trp1-Δ901 suc2-Δ9 lys2-801 GAL* |
| Strain, strain background (*S. cerevisiae*) | YAI4 | This study | | SEY6210 *natNT::PCUP1-GFP-FAR9* Stocked in T Kanki lab. |
| Strain, strain background (*S. cerevisiae*) | YAI19 | This study | | SEY6210 *FAR8-GFP::TRP1* Stocked in T Kanki lab. |
| Strain, strain background (*S. cerevisiae*) | YAI20 | This study | | SEY6210 *FAR11-GFP::TRP1* Stocked in T Kanki lab. |
| Strain, strain background (*S. cerevisiae*) | YAI21 | This study | | SEY6210 *FAR3-GFP::TRP1 SEC63-mCherry::hphNT* Stocked in T Kanki lab. |
| Strain, strain background (*S. cerevisiae*) | YAI22 | This study | | SEY6210 *FAR7-GFP::TRP1 SEC63-mCherry::hphNT* Stocked in T Kanki lab. |
| Strain, strain background (*S. cerevisiae*) | YAI23 | This study | | SEY6210 *FAR8-GFP::TRP1 SEC63-mCherry::hphNT* Stocked in T Kanki lab. |
| Strain, strain background (*S. cerevisiae*) | YAI24 | This study | | SEY6210 *natNT::PCUP1-GFP-FAR9 SEC63-mCherry::hphNT* Stocked in T Kanki lab. |
| Strain, strain background (*S. cerevisiae*) | YAI25 | This study | | SEY6210 *natNT::PCUP1-GFP-FAR10 SEC63-mCherry::hphNT* Stocked in T Kanki lab. |
| Strain, strain background (*S. cerevisiae*) | YAI26 | This study | | SEY6210 *FAR11-GFP::TRP1 SEC63-mCherry::hphNT* Stocked in T Kanki lab. |
| Strain, strain background (*S. cerevisiae*) | YKF179 | This study | | SEY6210 *far3::loxP-LEU2-loxP* Stocked in T Kanki lab. |
| Strain, strain background (*S. cerevisiae*) | YKF180 | This study | | SEY6210 *far7::loxP-LEU2-loxP* Stocked in T Kanki lab. |
| Strain, strain background (*S. cerevisiae*) | YKF181 | This study | | SEY6210 *far8::loxP-LEU2-loxP* Stocked in T Kanki lab. |
| Strain, strain background (*S. cerevisiae*) | YKF182 | This study | | SEY6210 *far9::loxP-LEU2-loxP* Stocked in T Kanki lab. |
| Strain, strain background (*S. cerevisiae*) | YKF183 | This study | | SEY6210 *far10::loxP-LEU2-loxP* Stocked in T Kanki lab. |
| Strain, strain background (*S. cerevisiae*) | YKF184 | This study | | SEY6210 *far11::loxP-LEU2-loxP* Stocked in T Kanki lab. |
| Strain, strain background (*S. cerevisiae*) | YAI27 | This study | | SEY6210 *natNT::PCUP1-GFP-FAR9ΔTA::TRP1* Stocked in T Kanki lab. |
| Strain, strain background (*S. cerevisiae*) | YAI28 | This study | | SEY6210 *natNT::PCUP1-GFP-FAR10ΔTA::TRP1* Stocked in T Kanki lab. |
| Strain, strain background (*S. cerevisiae*) | TKYM307 | DOI: 10.1016/j.celrep.2018.05.064 | | SEY6210 *IDH1-GFP::TRP1* |
| Strain, strain background (*S. cerevisiae*) | YKF170 | This study | | SEY6210 *IDH1-GFP::TRP1 FAR9ΔTA-3HA::HIS3* Stocked in T Kanki lab. |
| Strain, strain background (*S. cerevisiae*) | YKF172 | This study | | SEY6210 *IDH1-GFP::TRP1 far9::kanMX far10::hphNT* Stocked in T Kanki lab. |
| Strain, strain background (*S. cerevisiae*) | YKF175 | This study | | SEY6210 *IDH1-GFP::TRP1 FAR10ΔTA-3HA::kanMX* Stocked in T Kanki lab. |

*Continued on next page*

*Continued*

| Reagent type (species) or resource | Designation | Source or reference | Identifiers | Additional information |
|---|---|---|---|---|
| Strain, strain background (*S. cerevisiae*) | YKF176 | This study | | SEY6210 *IDH1-GFP::TRP1 FAR9ΔTA-3HA::HIS3 FAR10ΔTA-3HA::kanMX* Stocked in T Kanki lab. |
| Strain, strain background (*S. cerevisiae*) | YKF203 | This study | | SEY6210 *natNT::PCUP1-GFP-FAR9 SEC63-mCherry::hphNT far10::loxP-HIS3-loxP* Stocked in T Kanki lab. |
| Strain, strain background (*S. cerevisiae*) | YKF204 | This study | | SEY6210 *natNT::PCUP1-GFP-FAR9-TOM5$^{TA}$::TRP1 SEC63-mCherry::hphNT* Stocked in T Kanki lab. |
| Strain, strain background (*S. cerevisiae*) | YKF205 | This study | | SEY6210 *natNT::PCUP1-GFP-FAR9-TOM5$^{TA}$::TRP1 SEC63-mCherry::hphNT far10::loxP-HIS3-loxP* Stocked in T Kanki lab. |
| Strain, strain background (*S. cerevisiae*) | YKF206 | This study | | SEY6210 *natNT::PCUP1-GFP-FAR9-CYB5$^{TA}$::TRP1 SEC63-mCherry::hphNT* Stocked in T Kanki lab. |
| Strain, strain background (*S. cerevisiae*) | YKF207 | This study | | SEY6210 *natNT::PCUP1-GFP-FAR9-CYB5$^{TA}$::TRP1 SEC63-mCherry::hphNT far10::loxP-HIS3-loxP* Stocked in T Kanki lab. |
| Strain, strain background (*S. cerevisiae*) | YKF213 | This study | | SEY6210 *FAR3-GFP::TRP1 FAR9-TOM5$^{TA}$::HIS3* Stocked in T Kanki lab. |
| Strain, strain background (*S. cerevisiae*) | YKF214 | This study | | SEY6210 *FAR7-GFP::TRP1 FAR9-TOM5$^{TA}$::HIS3* Stocked in T Kanki lab. |
| Strain, strain background (*S. cerevisiae*) | YKF215 | This study | | SEY6210 *FAR8-GFP::TRP1 FAR9-TOM5$^{TA}$::HIS3* Stocked in T Kanki lab. |
| Strain, strain background (*S. cerevisiae*) | YKF216 | This study | | SEY6210 *natNT::PCUP1-GFP-FAR9-TOM5$^{TA}$::TRP1* Stocked in T Kanki lab. |
| Strain, strain background (*S. cerevisiae*) | YKF217 | This study | | SEY6210 *natNT::PCUP1-GFP-FAR10 FAR9-TOM5$^{TA}$::HIS3* Stocked in T Kanki lab. |
| Strain, strain background (*S. cerevisiae*) | YKF218 | This study | | SEY6210 *FAR11-GFP::TRP1 FAR9-TOM5$^{TA}$::HIS3* Stocked in T Kanki lab. |
| Strain, strain background (*S. cerevisiae*) | BY4741 | DOI:https://doi.org/10.1002/(SICI)1097-0061(19980130)14:2%3C115::AID-YEA204%3E3.0.CO;2-2 | | *MATa his3Δ1 leu2Δ0 met15Δ0 ura3Δ0* |
| Strain, strain background (*S. cerevisiae*) | BY *tsc11-1* | EUROSCARF (http://euroscarf.de/) | Y41093 | BY4741 *tsc11-1::kanMX* |
| Strain, strain background (*S. cerevisiae*) | YKF220 | This study | | BY *tsc11-1 natNT::PCUP1-GFP-FAR9* Stocked in T Kanki lab. |
| Strain, strain background (*S. cerevisiae*) | YKF221 | This study | | BY *tsc11-1 natNT::PCUP1-GFP-FAR9-TOM5$^{TA}$::TRP1* Stocked in T Kanki lab. |
| Strain, strain background (*S. cerevisiae*) | YKF222 | This study | | BY *tsc11-1 natNT::PCUP1-GFP-FAR9-CYB5$^{TA}$::TRP1* Stocked in T Kanki lab. |

*Continued on next page*

Continued

| Reagent type (species) or resource | Designation | Source or reference | Identifiers | Additional information |
|---|---|---|---|---|
| Strain, strain background (*S. cerevisiae*) | YKF223 | This study | | BY *tsc11-1 far9::loxP-HIS3-loxP* Stocked in T Kanki lab. |
| Strain, strain background (*S. cerevisiae*) | YKF224 | This study | | BY *tsc11-1 natNT::PCUP1-GFP-FAR9ΔTA::HIS3* Stocked in T Kanki lab. |
| Strain, strain background (*S. cerevisiae*) | TKYM80 | DOI: 10.1016/j.devcel.2009.06.014 | | SEY6210 *IDH1-GFP::TRP1 atg1::HIS3* |
| Strain, strain background (*S. cerevisiae*) | YKF29 | DOI: 10.1016/j.celrep.2018.05.064 | | SEY6210 *IDH1-GFP::TRP1 ppg1::kanMX* |
| Strain, strain background (*S. cerevisiae*) | YKF30 | DOI: 10.1016/j.celrep.2018.05.064 | | SEY6210 *IDH1-GFP::TRP1 natNT::PTEF-3HA-PPG1* |
| Strain, strain background (*S. cerevisiae*) | YKF76 | DOI: 10.1016/j.celrep.2018.05.064 | | SEY6210 *IDH1-GFP::TRP1 far9::kanMX* |
| Strain, strain background (*S. cerevisiae*) | YKF225 | This study | | SEY6210 *IDH1-GFP::TRP1 FAR9-TOM5$^{TA}$::HIS3* Stocked in T Kanki lab. |
| Strain, strain background (*S. cerevisiae*) | YKF226 | This study | | SEY6210 *IDH1-GFP::TRP1 FAR9-TOM5$^{TA}$::HIS3 ppg1::kanMX* Stocked in T Kanki lab. |
| Strain, strain background (*S. cerevisiae*) | YKF227 | This study | | SEY6210 *IDH1-GFP::TRP1 natNT::PTEF-3HA-PPG1 FAR9-TOM5$^{TA}$::HIS3* Stocked in T Kanki lab. |
| Strain, strain background (*S. cerevisiae*) | YKF228 | This study | | SEY6210 *IDH1-GFP::TRP1 FAR9-CYB5$^{TA}$::HIS3* Stocked in T Kanki lab. |
| Strain, strain background (*S. cerevisiae*) | YKF85 | This study | | SEY6210 *IDH1-GFP::TRP1 FAR3-3HA::HIS3* Stocked in T Kanki lab. |
| Strain, strain background (*S. cerevisiae*) | YKF86 | This study | | SEY6210 *IDH1-GFP::TRP1 FAR7-3HA::HIS3* Stocked in T Kanki lab. |
| Strain, strain background (*S. cerevisiae*) | YKF87 | This study | | SEY6210 *IDH1-GFP::TRP1 FAR8-3HA::HIS3* Stocked in T Kanki lab. |
| Strain, strain background (*S. cerevisiae*) | YKF90 | This study | | SEY6210 *IDH1-GFP::TRP1 FAR11-3HA::HIS3* Stocked in T Kanki lab. |
| Strain, strain background (*S. cerevisiae*) | YKF190 | This study | | SEY6210 *ppg1::kanMX far3::loxP far7::loxP far8::loxP far9::loxP-LEU2-loxP far10::loxP-HIS3-loxP far11::loxP* Stocked in T Kanki lab. |
| Strain, strain background (*S. cerevisiae*) | YKF191 | This study | | SEY6210 *ppg1::kanMX far3::loxP far7::loxP far8::loxP natNT::PCUP1-GFP-FAR9 far10::loxP-HIS3-loxP far11::loxP* Stocked in T Kanki lab. |
| Strain, strain background (*S. cerevisiae*) | YKF192 | This study | | SEY6210 *ppg1::kanMX far3::loxP far7::loxP far8::loxP far9::loxP-LEU2-loxP natNT::PCUP1-GFP-FAR10 far11::loxP* Stocked in T Kanki lab. |
| Strain, strain background (*S. cerevisiae*) | YKF193 | This study | | SEY6210 *far3::loxP-LEU2::loxP ppg1::kanMX* Stocked in T Kanki lab. |

*Continued on next page*

*Continued*

| Reagent type (species) or resource | Designation | Source or reference | Identifiers | Additional information |
|---|---|---|---|---|
| Strain, strain background (*S. cerevisiae*) | YKF194 | This study | | SEY6210 *far7::loxP-LEU2:: loxP ppg1::kanMX*<br>Stocked in T Kanki lab. |
| Strain, strain background (*S. cerevisiae*) | YKF195 | This study | | SEY6210 *far8::loxP-LEU2::loxP ppg1::kanMX*<br>Stocked in T Kanki lab. |
| Strain, strain background (*S. cerevisiae*) | YKF196 | This study | | SEY6210 *far11::loxP-LEU2::loxP ppg1::kanMX*<br>Stocked in T Kanki lab. |
| Strain, strain background (*S. cerevisiae*) | YKF197 | This study | | SEY6210 *natNT::PCUP1-GFP-FAR9 ppg1::kanMX*<br>Stocked in T Kanki lab. |
| Strain, strain background (*S. cerevisiae*) | YKF198 | This study | | SEY6210 *natNT::PCUP1-GFP-FAR10 ppg1::kanMX*<br>Stocked in T Kanki lab. |
| Strain, strain background (*S. cerevisiae*) | YKF200 | This study | | SEY6210 *TPD3-3HA::hphNT*<br>Stocked in T Kanki lab. |
| Strain, strain background (*S. cerevisiae*) | YKF230 | This study | | SEY6210 *TPD3-3HA::hphNT ppg1::kanMX*<br>Stocked in T Kanki lab. |
| Strain, strain background (*S. cerevisiae*) | YKF231 | This study | | SEY6210 *TPD3-3HA::hphNT ppg1::kanMX far11::loxP-LEU2-loxP*<br>Stocked in T Kanki lab. |
| Strain, strain background (*S. cerevisiae*) | YKF233 | This study | | SEY6210 *tpd3::natNT ppg1::kanMX*<br>Stocked in T Kanki lab. |
| Strain, strain background (*S. cerevisiae*) | YAI76 | This study | | SEY6210 *FAR3-3HA::hphNT FAR7-3HA::HIS3 natNT::PCUP1-GFP-FAR9*<br>Stocked in T Kanki lab. |
| Strain, strain background (*S. cerevisiae*) | YAI115 | This study | | SEY6210 *FAR3-3HA::hphNT FAR7-3HA::HIS3 natNT::PCUP1-GFP-FAR9 far8::loxP-LEU2-loxP*<br>Stocked in T Kanki lab. |
| Strain, strain background (*S. cerevisiae*) | YAI116 | This study | | SEY6210 *FAR3-3HA::hphNT FAR7-3HA::HIS3 natNT::PCUP1-GFP-FAR9 ppg1::kanMX*<br>Stocked in T Kanki lab. |
| Strain, strain background (*S. cerevisiae*) | BY4742 | DOI:https://doi.org/10.1002/(SICI)1097-0061(19980130)14:2%3C115::AID-YEA204%3E3.0.CO;2-2 | | *MATα his3Δ1 leu2Δ0 lys2Δ0 ura3Δ0* |
| Strain, strain background (*S. cerevisiae*) | BY *pph21Δ* | Thermo Fisher Scientific | 13831 | BY4742 *pph21::kanMX* |
| Strain, strain background (*S. cerevisiae*) | BY *pph22Δ* | Thermo Fisher Scientific | 13886 | BY4742 *pph22::kanMX* |
| Strain, strain background (*S. cerevisiae*) | BY *pph3Δ* | Thermo Fisher Scientific | 14010 | BY4742 *pph3::kanMX* |
| Strain, strain background (*S. cerevisiae*) | BY *sit4Δ* | Thermo Fisher Scientific | 13744 | BY4742 *sit4::kanMX* |
| Strain, strain background (*S. cerevisiae*) | BY *ppg1Δ* | Thermo Fisher Scientific | 15407 | BY4742 *ppg1::kanMX* |
| Strain, strain background (*S. cerevisiae*) | BY *far8Δ* | Thermo Fisher Scientific | 10604 | BY4742 *far8::kanMX* |
| Strain, strain background (*S. cerevisiae*) | BY *far11Δ* | Thermo Fisher Scientific | 12949 | BY4742 *far11::kanMX* |

*Continued*

| Reagent type (species) or resource | Designation | Source or reference | Identifiers | Additional information |
|---|---|---|---|---|
| Strain, strain background (*S. cerevisiae*) | YAI58 | This study | | SEY6210 *FAR3-GFP::TRP1 SEC63-mCherry::hphNT ppg1::kanMX* Stocked in T Kanki lab. |
| Strain, strain background (*S. cerevisiae*) | YAI59 | This study | | SEY6210 *FAR7-GFP::TRP1 SEC63-mCherry::hphNT ppg1::kanMX* Stocked in T Kanki lab. |
| Strain, strain background (*S. cerevisiae*) | YAI60 | This study | | SEY6210 *FAR8-GFP::TRP1 SEC63-mCherry::hphNT ppg1::kanMX* Stocked in T Kanki lab. |
| Strain, strain background (*S. cerevisiae*) | YAI61 | This study | | SEY6210 *natNT::PCUP1-GFP-FAR9 SEC63-mCherry::hphNT ppg1::kanMX* Stocked in T Kanki lab. |
| Strain, strain background (*S. cerevisiae*) | YAI62 | This study | | SEY6210 *natNT::PCUP1-GFP-FAR10 SEC63-mCherry::hphNT ppg1::kanMX* Stocked in T Kanki lab. |
| Strain, strain background (*S. cerevisiae*) | YAI63 | This study | | SEY6210 *FAR11-GFP::TRP1 SEC63-mCherry::hphNT ppg1::kanMX* Stocked in T Kanki lab. |
| Strain, strain background (*S. cerevisiae*) | YKF235 | This study | | SEY6210 *FAR11-GFP::TRP1 FAR9-TOM5$^{TA}$::HIS3 SEC63-mCherry::hphNT ppg1::kanMX* Stocked in T Kanki lab. |
| Strain, strain background (*S. cerevisiae*) | YKF26 | DOI: 10.1016/j.celrep.2018.05.064 | | SEY6210 *ppg1::kanMX* |
| Strain, strain background (*S. cerevisiae*) | TKYM139 | DOI: 10.1016/j.devcel.2009.06.014 | | SEY6210 *atg32::LEU2* |
| Strain, strain background (*S. cerevisiae*) | YKF57 | This study | | SEY6210 *atg32::LEU2 ppg1::kanMX* Stocked in T Kanki lab. |
| Strain, strain background (*S. cerevisiae*) | YAI70 | This study | | SEY6210 *atg32::LEU2 ppg1::kanMX far3::loxP-HIS3-loxP* Stocked in T Kanki lab. |
| Strain, strain background (*S. cerevisiae*) | YAI71 | This study | | SEY6210 *atg32::LEU2 ppg1::kanMX far7::loxP-HIS3-loxP* Stocked in T Kanki lab. |
| Strain, strain background (*S. cerevisiae*) | YAI72 | This study | | SEY6210 *atg32::LEU2 ppg1::kanMX far8::loxP-HIS3-loxP* Stocked in T Kanki lab. |
| Strain, strain background (*S. cerevisiae*) | YAI73 | This study | | SEY6210 *atg32::LEU2 ppg1::kanMX far9::loxP-HIS3-loxP* Stocked in T Kanki lab. |
| Strain, strain background (*S. cerevisiae*) | YAI74 | This study | | SEY6210 *atg32::LEU2 ppg1::kanMX far10::loxP-HIS3-loxP* Stocked in T Kanki lab. |
| Strain, strain background (*S. cerevisiae*) | YAI75 | This study | | SEY6210 *atg32::LEU2 ppg1::kanMX far11::loxP-HIS3-loxP* Stocked in T Kanki lab. |
| Strain, strain background (*S. cerevisiae*) | YKF35 | DOI: 10.1016/j.celrep.2018.05.064 | | SEY6210 *atg11::LEU2 atg32::HIS3 ppg1::kanMX* |
| Strain, strain background (*S. cerevisiae*) | YKF74 | DOI: 10.1016/j.celrep.2018.05.064 | | SEY6210 *IDH1-GFP::TRP1 far8::kanMX* |

*Continued on next page*

*Continued*

| Reagent type (species) or resource | Designation | Source or reference | Identifiers | Additional information |
|---|---|---|---|---|
| Strain, strain background (*S. cerevisiae*) | YKF160 | This study | | SEY6210 *IDH1-GFP::TRP1 far8::kanMX ppg1::natNT* Stocked in T Kanki lab. |
| Strain, strain background (*S. cerevisiae*) | YKF260 | This study | | SEY6210 *FAR11-GFP::TRP1 far8:: loxP-HIS3-loxP* Stocked in T Kanki lab. |
| Strain, strain background (*S. cerevisiae*) | TKYM312 | DOI: 10.1016/j.celrep.2018.05.064 | | SEY6210 *IDH1-GFP::TRP1 atg32::LEU2* |
| Strain, strain background (*S. cerevisiae*) | YKF261 | This study | | SEY6210 *IDH1-GFP::TRP1 atg32::LEU2 ppg1::kanMX* Stocked in T Kanki lab. |
| Strain, strain background (*S. cerevisiae*) | YKF262 | This study | | SEY6210 *IDH1-GFP::TRP1 atg32::LEU2 far8::kanMX* Stocked in T Kanki lab. |
| Strain, strain background (*S. cerevisiae*) | YKF263 | This study | | SEY6210 *IDH1-GFP::TRP1 atg32::LEU2 far8::kanMX ppg1::natNT* Stocked in T Kanki lab. |
| Strain, strain background (*S. cerevisiae*) | YKF264 | This study | | SEY6210 *IDH1-GFP::TRP1 atg32::LEU2 far8::kanMX far3::loxP-HIS3-loxP* Stocked in T Kanki lab. |
| Strain, strain background (*S. cerevisiae*) | YKF265 | This study | | SEY6210 *IDH1-GFP::TRP1 atg32::LEU2 far8::kanMX far7::loxP-HIS3-loxP* Stocked in T Kanki lab. |
| Strain, strain background (*S. cerevisiae*) | YKF266 | This study | | SEY6210 *IDH1-GFP::TRP1 atg32::LEU2 far8::kanMX far9::loxP-HIS3-loxP* Stocked in T Kanki lab. |
| Strain, strain background (*S. cerevisiae*) | YKF267 | This study | | SEY6210 *IDH1-GFP::TRP1 atg32::LEU2 far8::kanMX far10::loxP-HIS3-loxP* Stocked in T Kanki lab. |
| Strain, strain background (*S. cerevisiae*) | YKF268 | This study | | SEY6210 *IDH1-GFP::TRP1 atg32::LEU2 far8::kanMX far11::loxP-HIS3-loxP* Stocked in T Kanki lab. |
| Strain, strain background (*E. coli*) | BL21-CodonPlus (DE3)-RIL | Agilent | Cat# 230245 | B F⁻ *ompT hsdS*(r⁻$_B$ m⁻$_B$) *dcm*⁺ Tet$^r$ *gal* λ(DE3) *endA* Hte [*argU ileY leuW* Cam$^r$] |
| Antibody | anti-GFP (Mouse monoclonal) | Takara Bio | Cat# 632380, RRID:AB_10013427 | WB (1:5000) |
| Antibody | anti-HA (Mouse monoclonal) | Sigma-Aldrich | Cat# H9658, RRID:AB_260092 | WB (1:2500) |
| Antibody | anti-Pgk1 (Mouse monoclonal) | Thermo Fisher Scientific | Cat# 459250, RRID:AB_ 2532235 | WB (1:5000) |
| Antibody | anti-mouse IgG (Goat polyclonal, Peroxidase conjugated) | Merck Millipore | Cat# AP124P, RRID:AB_90456 | WB (1:10000) |
| Antibody | anti-rabbit IgG (Goat polyclonal, Peroxidase conjugated) | Jackson ImmunoResearch | Cat# 111-035-003, RRID:AB_2313567 | WB (1:10000) |

*Continued*

| Reagent type (species) or resource | Designation | Source or reference | Identifiers | Additional information |
|---|---|---|---|---|
| Antibody | anti-Atg32 (Rabbit polyclonal) | DOI: 10.1091/mbc.E11-02-0145 | | WB (1:2500) |
| Antibody | anti-Ppg1 (Rabbit polyclonal) | This study | | WB (1:1000) Stocked in T Kanki lab. |
| Antibody | anti-Far8 (Rabbit polyclonal) | This study | | WB (1:1000) Stocked in T Kanki lab. |
| Antibody | anti-Far9 (Rabbit polyclonal) | This study | | WB (1:1000) Stocked in T Kanki lab. |
| Antibody | anti-Far11 (Rabbit polyclonal) | This study | | WB (1:1000) Stocked in T Kanki lab. |
| Recombinant DNA reagent | pCu416 (plasmid) | DOI: 10.1016/s0076-6879(99)06010-3 | | *CEN/ARS URA3 PCUP1* |
| Recombinant DNA reagent | pCu416-PPG1 (plasmid) | DOI: 10.1016/j.celrep.2018.05.064 | | *CEN/ARS URA3 PCUP1-PPG1* |
| Recombinant DNA reagent | pCu416-FLAG-His6-PPG1 (plasmid) | DOI: 10.1016/j.celrep.2018.05.064 | | *CEN/ARS URA3 PCUP1-FLAG-His6-PPG1* |
| Recombinant DNA reagent | pCu416-FLAG-His6-PPG1$^{H111N}$ (plasmid) | This study | | *CEN/ARS URA3 PCUP1-FLAG-His6-PPG1$^{H111N}$* Stocked in T Kanki lab. |
| Recombinant DNA reagent | pCu414-FAR3-3HA (plasmid) | This study | | *CEN/ARS TRP1 PCUP1-FAR3-3HA* Stocked in T Kanki lab. |
| Recombinant DNA reagent | pCu414-FAR7-3HA (plasmid) | This study | | *CEN/ARS TRP1 PCUP1-FAR7-3HA* Stocked in T Kanki lab. |
| Recombinant DNA reagent | pCu414-FAR8-3HA (plasmid) | This study | | *CEN/ARS TRP1 PCUP1-FAR8-3HA* Stocked in T Kanki lab. |
| Recombinant DNA reagent | pCu414-FAR11-3HA (plasmid) | This study | | *CEN/ARS TRP1 PCUP1-FAR11-3HA* Stocked in T Kanki lab. |
| Recombinant DNA reagent | pCu416-3HA-ATG32 (plasmid) | This study | | *CEN/ARS URA3 PCUP1-3HA-ATG32* Stocked in T Kanki lab. |
| Recombinant DNA reagent | pCu416-3HA-ATG32-2SA (plasmid) | This study | | *CEN/ARS URA3 PCUP1-3HA-ATG32$^{S114A/S119A}$* Stocked in T Kanki lab. |
| Recombinant DNA reagent | pCu416-3HA-ATG32Δ151–200 (plasmid) | This study | | *CEN/ARS URA3 PCUP1-3HA-ATG32Δ151–200* Stocked in T Kanki lab. |
| Recombinant DNA reagent | pCu416-ATG32 (plasmid) | DOI: 10.1091/mbc.E11-02-0145 | | *CEN/ARS URA3 PCUP1-ATG32* |
| Recombinant DNA reagent | pCu416-FAR8-ATG32 (plasmid) | This study | | *CEN/ARS URA3 PCUP1-FAR8-ATG32* Stocked in T Kanki lab. |
| Recombinant DNA reagent | pRS416-ATG32 (plasmid) | DOI: 10.1016/j.devcel.2009.06.014 | | *CEN/ARS URA3 PATG32-ATG32* |
| Recombinant DNA reagent | pRS416-FAR8-ATG32 (plasmid) | This study | | *CEN/ARS URA3 PATG32-FAR8-ATG32* Stocked in T Kanki lab. |

*Continued on next page*

*Continued*

| Reagent type (species) or resource | Designation | Source or reference | Identifiers | Additional information |
|---|---|---|---|---|
| Recombinant DNA reagent | pPROEX-HTb (plasmid) | Invitrogen | Cat# 10711018 | *Amp$^r$ lacI$^q$ Ptrc-His6* |
| Recombinant DNA reagent | pPROEX-FAR8 (plasmid) | This study | | *Amp$^r$ lacI$^q$ Ptrc-His6-FAR8* Stocked in T Kanki lab. |
| Recombinant DNA reagent | pGEX-4T-1 (plasmid) | GE Healthcare | Cat# 28954549 | *Amp$^r$ lacI$^q$ Ptac-GST* |
| Recombinant DNA reagent | pGEX-ATG32(N250) (plasmid) | DOI: 10.1091/mbc.E11-02-0145 | | *Amp$^r$ lacI$^q$ Ptac-GST-ATG32(N250)* |
| Recombinant DNA reagent | pGEX-ATG32(N200) (plasmid) | This study | | *Amp$^r$ lacI$^q$ Ptac-GST-ATG32(N200)* Stocked in T Kanki lab. |
| Recombinant DNA reagent | pGEX-ATG32(N150) (plasmid) | This study | | *Amp$^r$ lacI$^q$ Ptac-GST-ATG32(N150)* Stocked in T Kanki lab. |
| Commercial assay or kit | EzWestLumi plus | Atto | Cat# WSE-7120 | |
| Commercial assay or kit | Clarity Max Western ECL Substrate | Bio-Rad | Cat# 1705062 | |
| Commercial assay or kit | anti-FLAG M2 affinity gel | Sigma-Aldrich | Cat# A2220 | |
| Commercial assay or kit | Protein G Sepharose 4 Fast Flow | GE Healthcare | Cat# 17061801 | |
| Commercial assay or kit | Ni Sepharose 6 Fast Flow | GE Healthcare | Cat# 17531801 | |
| Commercial assay or kit | Glutathione Sepharose 4 Fast Flow | GE Healthcare | Cat# 17075601 | |
| Chemical compound, drug | MitoTracker Red CMXRos | Thermo Fisher Scientific | Cat# M7512 | (50 nM) |
| Chemical compound, drug | Rapamycin | LC Laboratories | Cat# R-5000 | (100 nM) |
| Software, algorithm | Image Lab | Bio-Rad | | |
| Software, algorithm | MetaMorph 7 | Molecular Devices | | |

## Yeast strains

The yeast strains used in this study are shown in Key Resources Table. Gene deletion and tagging were performed as described previously (*Gueldener et al., 2002*; *Janke et al., 2004*; *Longtine et al., 1998*).

## Growth media

Yeast cells were cultured at 30°C in rich medium (YPD: 1% yeast extract, 2% peptone, and 2% glucose), lactate medium (YPL; 1% yeast extract, 2% peptone, and 2% lactate), or synthetic minimal medium with glucose (SMD; 0.67% yeast nitrogen base, 2% glucose, and amino acids) or lactate (SML; 0.67% yeast nitrogen base, 2% lactate, and amino acids). Nitrogen starvation experiments were performed in synthetic minimal medium lacking nitrogen (SD-N; 0.17% yeast nitrogen base without amino acids and ammonium sulfate and 2% glucose). For temperature-sensitive assays (repeated independently three times), serially diluted cells were spotted on YPD agar plates and cultured at 30°C or 37°C.

## Plasmids

The plasmids used in this study are shown in Key Resources Table. To construct a C-terminally 3HA-tagged Far3 expression plasmid, a DNA fragment encoding the *FAR3-3HA* gene with *Spe*I and *Xho*I sites was amplified from the genomic DNA of the *FAR3-3HA* strain and inserted into the same sites

of pCu414 (*Labbé and Thiele, 1999*). Far7-3HA, Far8-3HA, and Far11-3HA expression plasmids were constructed using the same method. To construct an N-terminally 3HA-tagged Atg32 expression plasmid under the control of the *CUP1* promoter, a DNA fragment encoding the *ATG32* gene with *Eco*RI and *Sal*I sites was amplified by polymerase chain reaction (PCR) from the yeast genomic DNA and inserted into the same sites of pCu3HA(416) (*Wang et al., 2012*). The derivatives of 3HA-Atg32 expression plasmids (2SA, 2SD, and Δ151–200) and the *PPG1-H111N* mutation were generated by PCR-mediated mutagenesis. To construct an N-terminally His6-tagged Far8 expression plasmid, a DNA fragment encoding the *FAR8* gene with *Bam*HI and *Xho*I sites was amplified by PCR from the yeast genomic DNA and inserted into the same sites of pPROEX-HTb (Invitrogen). The derivatives of N-terminally GST-tagged Atg32 expression plasmid were constructed by insertion of the *Bgl*II-*Sal*I fragments of *ATG32* with different lengths into the *Bam*HI-*Sal*I sites of pGEX-4T-1 (GE Healthcare). Far8-Atg32 chimeric protein expression plasmids were constructed by the insertion of the *Spe*I-*Sma*I fragment of *FAR8* into the same sites of pCu416-ATG32 (*Aoki et al., 2011*) or pRS416-ATG32 (*Kanki et al., 2009*).

### Antibodies

Anti-Ppg1, anti-Far8, anti-Far9, and anti-Far11 antibodies were produced by immunizing rabbits injected with the recombinant His6-tagged Ppg1, Far8, Far9, and Far11 proteins, respectively, and affinity purifying the serum using recombinant protein-conjugated Sepharose.

### Mitophagy assay

To monitor the mitophagy levels, an Idh1-GFP processing assay was performed (*Kanki and Klionsky, 2008*). The cells were cultured in YPD or SMD medium until the early log phase, and the cells were then shifted to YPL or SML medium (starting at $OD_{600} = 0.2$). The cells were collected after 20 or 24 hr (growing phase) and 40 or 48 hr (stationary phase), and cell lysates equivalent to $OD_{600} = 0.2$ units of cells were subjected to Western blot (WB) analysis using anti-GFP and anti-Atg32 antibodies.

### Western blot analysis

Protein samples from yeast cells were resuspended in sodium dodecyl sulfate (SDS) sampling buffer (50 mM Tris-HCl [pH 6.8], 10% glycerol, 2% SDS, 5% 2-mercaptoethanol, and 0.1% bromophenol blue), incubated at 42°C for 60 min, and subjected to SDS-polyacrylamide gel electrophoresis (PAGE). Proteins were transferred from polyacrylamide gels to polyvinylidene difluoride membranes (Merck Millipore) using transfer buffer (25 mM Tris, 192 mM glycine, 20% methanol). The membranes were blocked with phosphate-buffered saline (PBS) with Tween-20 (PBS-T; 10 mM $PO_4^{3-}$, 140 mM NaCl, 2.7 mM KCl, and 0.05% Tween-20) containing 5% skim milk for 1 hr. The membranes were incubated with primary antibodies in PBS-T containing 2% skim milk overnight at 4°C and washed three times with PBS-T. The membranes were then incubated with secondary antibodies in PBS-T containing 2% skim milk for 1 hr at room temperature and washed three times with PBS-T. Chemiluminescence signals were detected using ChemiDoc XRS+ (Bio-Rad) and analyzed using Image Lab software (Bio-Rad). The WB experiments, including Atg32 phosphorylation, mitophagy, immunoprecipitation, and pull-down assays, were independently repeated at least three times (indicated in each figure legend).

### Fluorescence microscopy

Cells expressing GFP-fused Far proteins or derivatives were cultured in YPD medium until the early log growth phase. To stain the mitochondria, cells were incubated with 50 nM MitoTracker Red CMXRos (Thermo Fisher Scientific) for 30 min. Fluorescence signals were visualized using a microscope (IX73, Olympus) with a UPlanSApo 100 × oil objective lens and a cooled charge-coupled device camera (EXi Blue, QImaging). Fluorescence images were analyzed using MetaMorph seven software (Molecular Devices). More than 100 cells were analyzed in each microscopy experiment (independently repeated three times), and most of the analyzed cells showed the same localization pattern, as shown in each panel.

## Immunoprecipitation

Yeast cells were cultured in YPD or SMD medium until the early log growth phase. Then, 30 to 50 $OD_{600}$ units of cells were collected and frozen until use. The cells were lysed with glass beads in lysis buffer (PBS, 0.2% Triton X-100, 1 mM phenylmethylsulfonyl fluoride [PMSF], and Complete EDTA-free protease inhibitor [Roche]), and then centrifuged at 20,000 × g for 10 min at 4°C. For the immunoprecipitation of FLAG-His6-Ppg1, the supernatant was mixed with an anti-FLAG M2 affinity gel (Sigma-Aldrich) at 4°C for 4 hr. For the immunoprecipitation of Far8 or 3HA-Atg32, the supernatant was mixed with an anti-Far8 or anti-HA (Sigma-Aldrich) antibody at 4°C for 2 hr followed by the addition of Protein G Sepharose (GE Healthcare) for an additional 2 hr. The protein-bound beads were washed with lysis buffer five times, and the sample was eluted by an SDS sampling buffer. The elution samples were analyzed by WB.

## Protein expression and purification

*E. coli* BL21 (DE3) CodonPlus cells (Agilent) were used for the expression of His6-Far8, GST, and GST-tagged Atg32 (three derivatives) proteins. The cells were cultured in Luria-Bertani medium containing 100 µg/ml ampicillin until $OD_{600}$ = 0.6–1.0. Protein expression was then induced by adding 1 mM isopropyl β-D-thiogalactopyranoside at 30°C for 6 hr. The cells were harvested by centrifugation and stored at −80°C. Harvested *E. coli* cells were resuspended in lysis buffer (PBS, 1% Triton X-100, 1 mM PMSF, and Complete EDTA-free protease inhibitor) and disrupted by six cycles of sonication (30 s). The supernatant (soluble fraction) was separated from cell lysates by centrifugation at 20,000 × g at 4°C for 10 min. The soluble fractions for His6-Far8 and GST/GST-Atg32 were applied to Ni Sepharose (GE Healthcare) and glutathione Sepharose (GE Healthcare) at 4°C for 1 hr, respectively. After washing with the lysis buffer, His6-Far8 proteins were eluted with the same buffer containing 500 mM imidazole. The GST/GST-Atg32-bound glutathione Sepharose was washed and further used for the GST pull-down assay.

## GST pull-down assay

GST/GST-Atg32-bound glutathione Sepharose and His6-Far8 were mixed at 4°C for 1 hr in PBS with 0.2% Triton X-100. The beads were washed four times with the same buffer. Proteins were eluted by boiling with SDS sampling buffer, loaded on an SDS-PAGE gel, and detected by Coomassie brilliant blue (CBB) staining or WB with an anti-Far8 antibody.

## Acknowledgements

We thank Kyosuke Sakai and Kazuki Nishimizu for their support in the localization analyses of the Far complex.

## Additional information

### Funding

| Funder | Grant reference number | Author |
| --- | --- | --- |
| Japan Society for the Promotion of Science | 19K22419 | Tomotake Kanki |
| Japan Society for the Promotion of Science | 19H05712 | Tomotake Kanki |
| Japan Society for the Promotion of Science | 18H04858 | Tomotake Kanki |
| Japan Society for the Promotion of Science | 18H04691 | Tomotake Kanki |
| Japan Society for the Promotion of Science | 17H03671 | Tomotake Kanki |
| Japan Society for the Promotion of Science | 18K06129 | Kentaro Furukawa |
| Japan Agency for Medical Re- | JP18gm6110013h0001 | Tomotake Kanki |

| | | |
|---|---|---|
| search and Development | | |
| Takeda Science Foundation | | Kentaro Furukawa Tomoyuki Fukuda |
| Noda Institute for Scientific Research | | Kentaro Furukawa |
| Institute for Fermentation, Osaka | | Kentaro Furukawa |
| Niigata University | Kyowakai Medical Research Grant | Aleksei Innokentev |

The funders had no role in study design, data collection and interpretation, or the decision to submit the work for publication.

## Author contributions

Aleksei Innokentev, Formal analysis, Investigation, Visualization; Kentaro Furukawa, Conceptualization, Formal analysis, Supervision, Funding acquisition, Investigation, Visualization, Writing - original draft, Writing - review and editing; Tomoyuki Fukuda, Formal analysis, Funding acquisition, Investigation, Writing - review and editing; Tetsu Saigusa, Keiichi Inoue, Shun-ichi Yamashita, Formal analysis, Investigation; Tomotake Kanki, Conceptualization, Formal analysis, Supervision, Funding acquisition, Writing - original draft, Writing - review and editing

## Author ORCIDs

Kentaro Furukawa (iD) https://orcid.org/0000-0002-1551-1538
Tomoyuki Fukuda (iD) http://orcid.org/0000-0003-2069-7127
Tomotake Kanki (iD) https://orcid.org/0000-0001-9646-5379

## Decision letter and Author response

Decision letter https://doi.org/10.7554/eLife.63694.sa1
Author response https://doi.org/10.7554/eLife.63694.sa2

# Additional files

## Supplementary files

• Transparent reporting form

## Data availability

All data generated or analysed during this study are included in the manuscript and supporting files.

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
