## [Decision Letter]

**Acceptance summary:**

In this study, Innokentev et al. discovered that the Far complex, which regulates the protein phosphatase Ppg1, localizes to both mitochondria and the ER, and that these differently localized complexes play distinct roles, the regulation of mitophagy and the TORC2 pathway, respectively. The authors further clarified the mechanisms underlying the assembly of the Far complex and its association with the mitophagy receptor Atg32. This study will provide significant insights into our understanding of mitophagy regulation.

**Decision letter after peer review:**

[Editors’ note: the authors submitted for reconsideration following the decision after peer review. What follows is the decision letter after the first round of review.]

Thank you for submitting your work entitled "Association and dissociation between the mitochondrial Far complex and Atg32 regulates mitophagy" for consideration by *eLife*. Your article has been reviewed by three peer reviewers, one of whom is a member of our Board of Reviewing Editors, and the evaluation has been overseen by a Senior Editor. The reviewers have opted to remain anonymous.

Our decision has been reached after consultation between the reviewers, and summarized by the Reviewing Editor as you can see below. All the reviewers recognize the significance of this study that addresses an important mechanistic issue in mitophagy regulation. At the same time, the consensus amongst us is that two key conclusions are not convincingly supported by experiments in the present manuscript. Based on these discussions, we regret to inform you that your work will not be considered further for publication in *eLife*. However, in the future we will be happy to re-consider a new manuscript that addresses the issues outlined here.

Summary:

In earlier work, this group showed that the phosphatase Ppg1 and the Far complex cooperate to dephosphorylate the mitophagy receptor Atg32, resulting in mitophagy inhibition. In the present study, they take a closer look at this mechanism and the role of the Far complex. The authors discovered that a significant proportion of the Far complex localizes to mitochondria in addition to its previously-described ER localization. They then elegantly showed that these differently localized complexes play distinct roles, the regulation of mitophagy and the TORC2 pathway, respectively. The authors further clarified the mechanisms underlying the assembly of the Far complex and its association with Atg32, and how these processes are affected before and after mitophagy induction. Thus, this study will provide significant insights into our understanding of the regulation of mitophagy. However, the authors should perform further experiments to convincingly draw their conclusions.

Essential revisions

1) Two important discoveries in this manuscript, the regulation of Atg32-Far interaction and the dependence of Far-Ppg1 complex assembly on the phosphatase activity of Ppg1, are not fully validated and corroborated. The former would be the most important conclusion of this study and is highlighted by the authors in the title and the Abstract. However, the conclusion is based on a single piece of data in Figure 5G, and the quality of the immunoblotting image was not clear enough, which should be replaced with a better one. We also request the authors to provide additional evidence to strengthen this conclusion. For example, the authors should examine if the same result is obtained when mitophagy is induced in non-fermentable medium or rapamycin treatment instead of nitrogen starvation. In addition, it would be worth examining whether artificial tethering of a Far protein to Atg32 prevents the activation of mitophagy. As for the latter, it would be useful to examine whether the Ppg1-H111N mutant can interact normally with Far11 and Far8, respectively.

2) The assembly of the Far complex and Ppg1: The authors write: "We next investigated whether the Far complex interacts directly with Atg32". Whereas I agree that this is an important question to address, the authors do not address the directness of the interaction with their experiments. They solely use immunoprecipitations, with which distinguishing between a direct and an indirect interaction is not possible. As the authors draw a clear assembly model in the end, which is based on – possibly indirect – co-immunoprecipitation studies only, further experiments addressing the directness of these interactions are required to draw this conclusion. In general, this part on the Far complex and its (possibly different) assembly during TORC2 signaling and during mitophagy is the weakest part of the story and would profit of further analyses.

3) Statistics and reproducibility: For almost all experiments one single Western blot or one microscopy image only are shown. Some indication of reproducibility is needed. Whereas quantifications would be the best, at least a statement on how many times the experiment has been performed with the same outcome and how many cells out of how many cells showed the respective phenotype, is absolutely needed, at least in the figure legend.

Reviewer #1:

In this study, Kanki's group, by extending their previous findings, has advanced understanding of the regulatory mechanism of mitophagy in *S. cerevisiae*. Specifically, the authors discovered that a significant proportion of the Ppg1-Far protein complex, which dephosphorylates the mitophagy receptor Atg32 under mitophagy-suppressing conditions, localizes to mitochondria in addition to its previously-described ER localization, and then elegantly revealed that these differently localized complexes play distinct roles, the regulation of mitophagy and the TORC2 pathway, respectively. The authors further clarified the mechanisms underlying the assembly of the Far complex and its association with Atg32, and how these processes are affected before and after mitophagy induction. Consequently, this study has provided significant mechanistic insights into the regulation of mitophagy, in which the interaction between the Far complex and Atg32, rather than the assembly or localization of the Ppg1-Far complex, is regulated in response to mitophagy-inducing signals.

1) Figure 5G: This is the most important experiment to explain mitophagy regulation based on an alteration in the interaction between the Far complex and Atg32, but the quality of the immunoblotting image was not clear enough; should be replaced with a better one. In addition, the authors should examine if the same result is obtained when mitophagy is induced in non-fermentable medium or rapamycin treatment instead of nitrogen starvation.

2) Figure 4A: A "no antibody" control should be added to confirm that Far proteins other than Far11 were also coprecipitated with Far8 in a specific manner.

3) Figure 5E: It would be interesting to examine whether the S114D/S119D mutant showed a stronger association with the Far complex than the wild type protein. The results may strengthen the authors' model that the Far complex preferentially interacts with phosphorylated Atg32.

Reviewer #2:

This manuscript addresses the regulation of mitophagy by the Ppg1 phosphatase and the Far complex. In earlier work in 2018, the group showed that the phosphatase Ppg1 and the Far complex cooperate in counteracting Atg32 dependent mitophagy. This counteraction happens by dephosphorylating Atg32, which results in mitophagy inhibition.

In this new study they take a closer look at this mechanism and the role of the Far complex. They find that several Far complex members not only localize to the known location at the ER but also to mitochondria. A forced ER localization of the Far complex results in no inhibition of mitophagy, whereas a forced mitochondrial localization does, suggesting that the Far complex plays different roles, at the ER in TORC2 signaling, and at the mitochondria in mitophagy.

Further detailed analysis reveals insight into the Far complex assembly at mitochondria and its interaction with Atg32.

This is an elegant mechanistic study of known factors in autophagy. The field of autophagy largely lacks knowledge of mechanisms, many factors are known, but we don't know what they do. Such studies are important to move the field forward. In my opinion the experiments shown are very well done and only little revision is needed for the shown experiments.

I have only two additional comments:

1) The assembly of the Far complex and Ppg1: The authors write: "We next investigated whether the Far complex interacts directly with Atg32". Whereas I agree that this is an important question to address, the authors do not address the directness of the interaction with their experiments. They solely use immunoprecipitations, with which distinguishing between a direct and an indirect interaction is not possible. As the authors draw a clear assembly model in the end, which is based on – possibly indirect – co-immunoprecipitation studies only, further experiments addressing the directness of these interactions are required to draw this conclusion. In general, this part on the Far complex and its (possibly different) assembly during TORC2 signaling and during mitophagy is the weakest part of the story and would profit of further analyses.

2) Statistics and reproducibility: For almost all experiments one single Western blot or one microscopy image only are shown. Some indication of reproducibility is needed. Whereas quantifications would be the best, at least a statement on how many times the experiment has been performed with the same outcome and how many cells out of how many cells showed the respective phenotype, is absolutely needed, at least in the figure legend.

Reviewer #3:

In this manuscript, the authors followed up on their previous discovery that budding yeast mitophagy receptor Atg32 is negatively regulated by the Far-Ppg1 phosphatase complex, and dissected the subcellular localization, the complex composition, and the regulatory mechanisms of the Far-Ppg1 phosphatase complex. They revealed that the Far-Ppg1 phosphatase complex in budding yeast has a previously unknown mitochondrial localization and mitochondrial localized Far-Ppg1 is responsible for Atg32 inhibition. They found that Ppg1 is important for the assembly of Far11 with the rest of the Far proteins. They showed that Far8 interacts with Atg32 and this interaction is weakened under a mitophgay-inducing condition. These findings provide significant new insights into how Far-Ppg1 phosphatase acts, and enhance our understanding on how mitophagy is regulated.

I have the following suggestions for the authors to consider:

1) Two important discoveries in this manuscript, the regulation of Atg32-Far interaction and the dependence of Far-Ppg1 complex assembly on the phosphatase activity of Ppg1, seemed to me not fully validated and corroborated, and felt somewhat preliminary. The first under-validated conclusion is that Atg32 is activated by weakening the Atg32-Far interaction. This is probably the most important conclusion of this manuscript and is highlighted by the authors in the title and the Abstract. However, the conclusion is based on a single piece of data in Figure 5G. It would be useful to add additional evidence to strengthen this conclusion. For example, does artificial tethering of a Far protein to Atg32 prevent the activation of mitophagy?

2) The second under-validated conclusion is also only supported by a single piece of data in Figure 4E. It would be useful to examine whether the Ppg1-H111N mutant can interact normally with Far11 and Far8, respectively.

[Editors’ note: further revisions were suggested prior to acceptance, as described below.]

Thank you for resubmitting your work entitled "Association and dissociation between the mitochondrial Far complex and Atg32 regulate mitophagy" for further consideration by *eLife*. Your revised article has been evaluated by David Ron as the Senior Editor and a Reviewing Editor.

The manuscript has been improved but there are some remaining issues that need to be addressed before acceptance, as outlined below:

Summary:

This study by Innokentev et al. describes the mechanism of mitophagy regulation by the Ppg1-Far phosphatase complex in yeast. This is a resubmitted manuscript and has been assessed by the same reviewers who reviewed the original manuscript. The previous manuscript was rejected because the two major conclusions were not convincingly supported by experiments. We find that the authors have satisfactorily addressed these important issues but others remain.

Essential revisions

1) Statistics and reproducibility

The authors have not provided sufficient statistical details to enable the reader to gage the reproducibility of the observations reported, the new manuscript merely provides a general statement in the Materials and method section that the experiments were performed "two to three times". The key experiments should have been performed at least three times with the same outcome. And the reader must be made to know how many times each individual experiments was repeated, general statements are not as helpful. Presently information on repetitions is only mentioned in the Materials and methods section and generalized: " The WB experiments, including Atg32 phosphorylation, mitophagy, immunoprecipitation, and pull-down assays, were repeated independently twice or thrice." Especially for figures that report on less clear observations, such as Figures 3B, 3I, and 5G, it is important that the observations reported had been were reproduced at least three times and to commit to this unambiguously in each figure legend.

2) The “direct” interaction between the Far complex and Atg32

This has been well addressed by using recombinant proteins, showing that Far8 interacts directly with Atg32. Also the model has been adapted accordingly. These experiments clarify that the interaction between Atg32 and Far8 is direct. It seems however, that the interaction is way substoichiometric, as the interaction cannot be seen by coomassie staining for Far8? Why is this interaction so weak? How do the authors envision this works? This point should be addressed at least in the Discussion.

---

## [Author Response]

[Editors’ note: the authors resubmitted a revised version of the paper for consideration. What follows is the authors’ response to the first round of review.]

Essential revisions1) Two important discoveries in this manuscript, the regulation of Atg32-Far interaction and the dependence of Far-Ppg1 complex assembly on the phosphatase activity of Ppg1, are not fully validated and corroborated. The former would be the most important conclusion of this study and is highlighted by the authors in the title and the Abstract. However, the conclusion is based on a single piece of data in Figure 5G, and the quality of the immunoblotting image was not clear enough, which should be replaced with a better one. We also request the authors to provide additional evidence to strengthen this conclusion. For example, the authors should examine if the same result is obtained when mitophagy is induced in non-fermentable medium or rapamycin treatment instead of nitrogen starvation. In addition, it would be worth examining whether artificial tethering of a Far protein to Atg32 prevents the activation of mitophagy. As for the latter, it would be useful to examine whether the Ppg1-H111N mutant can interact normally with Far11 and Far8, respectively.

Following the first part of the comment, we reproduced the dissociation between Atg32 and Far8 upon nitrogen starvation and showed the result in Figure 5G. We believe that the current immunoblotting is clearer than the previous one.

Following the second part of the comment, we obtained additional data regarding dissociation between Atg32 and Far8 upon rapamycin treatment (Figure 5—figure supplement 2), supporting our conclusion that the interaction between Atg32 and Far8 is weakened by mitophagy-inducing conditions.

Following the third part of the comment, we expressed a Far8-Atg32 chimeric protein as artificial tethering of the Far complex to Atg32 and tested whether it affects the phosphorylation status of Atg32 and mitophagy. This Far8-Atg32 chimeric protein showed an unphosphorylated form even without endogenous Far8 under growing conditions, and it was not phosphorylated even under mitophagy-inducing conditions (Figures 6C and F). Moreover, the cells expressing the Far8-Atg32 chimeric protein showed impaired mitophagy (Idh1-GFP processing; Figure 6F). These effects of the Far8-Atg32 chimera protein were canceled by *PPG1* deletion. These results strongly support our conclusion that the association and dissociation between Atg32 and Far8 (Far complex) are crucial for mitophagy regulation.

We also found that Far3, Far7, Far9, and Far11 are still required for Far8-Atg32 dephosphorylation (Figure 6D) and that cells expressing Far8-Atg32, but not Atg32, recruit Far11 (Far11-GFP) to the mitochondria (Figure 6E). These results suggest that the Atg32-Far8 chimeric protein normally serves as a Far complex component and counteracts with the phosphorylation of the Atg32 part.

Following the last part of the comment, we examined whether Ppg1-H111N can interact with Far8 and Far11. We found that Ppg1-H111N can interact with Far11 and Far8, but the interaction is impaired (Figure 4G). This result suggests that the phosphatase activity of Ppg1 is required for the assembling integrity of Ppg1-Far11-Far8.

2) The assembly of the Far complex and Ppg1: The authors write: "We next investigated whether the Far complex interacts directly with Atg32". Whereas I agree that this is an important question to address, the authors do not address the directness of the interaction with their experiments. They solely use immunoprecipitations, with which distinguishing between a direct and an indirect interaction is not possible. As the authors draw a clear assembly model in the end, which is based on – possibly indirect – co-immunoprecipitation studies only, further experiments addressing the directness of these interactions are required to draw this conclusion. In general, this part on the Far complex and its (possibly different) assembly during TORC2 signaling and during mitophagy is the weakest part of the story and would profit of further analyses.

We agree with the reviewer that our previous manuscript did not include critical evidence for the direct interaction between Atg32 and the Far complex. To investigate whether the Far complex directly interacts with Atg32, we performed a GST pull-down assay using recombinant His6-Far8 and GST-Atg32 proteins produced in *E. coli*. As shown in Figure 6, GST-Atg32(N250), but not GST only or GST-Atg32(N200/N150), pulled down His6-Far8. This result, together with Figure 5E, indicates that Far8 directly interacts with Atg32 and that the 151–250 region of Atg32 is required for the interaction. Based on this finding, we modified our proposed model that Far8 interacts with Atg32 in Figure 7.

3) Statistics and reproducibility: For almost all experiments one single Western blot or one microscopy image only are shown. Some indication of reproducibility is needed. Whereas quantifications would be the best, at least a statement on how many times the experiment has been performed with the same outcome and how many cells out of how many cells showed the respective phenotype, is absolutely needed, at least in the figure legend.

We employed representative images from at least two to three independent experiments (yeast growth assay and Western blot, including Atg32 phosphorylation, mitophagy, immunoprecipitation, and pull-down assays). We mentioned this information in Materials and methods. In all microscopy experiments (two to three independent experiments analyzing 100–200 cells), most of the analyzed cells showed the same localization pattern, as shown in each panel. We mentioned this information in the main text and Materials and methods.

[Editors’ note: what follows is the authors’ response to the second round of review.]

Essential revisions1) Statistics and reproducibilityThe authors have not provided sufficient statistical details to enable the reader to gage the reproducibility of the observations reported, the new manuscript merely provides a general statement in the Materials and method section that the experiments were performed "two to three times". The key experiments should have been performed at least three times with the same outcome. And the reader must be made to know how many times each individual experiments was repeated, general statements are not as helpful. Presently information on repetitions is only mentioned in the Materials and methods section and generalized: " The WB experiments, including Atg32 phosphorylation, mitophagy, immunoprecipitation, and pull-down assays, were repeated independently twice or thrice." Especially for figures that report on less clear observations, such as Figures 3B, 3I, and 5G, it is important that the observations reported had been were reproduced at least three times and to commit to this unambiguously in each figure legend.

We apologize for not providing sufficient details of the experimental reproducibility. We repeated all of the experiments that were performed only two times in the original manuscript, and all of the repeated experiments gave the same outcome. We mentioned how many times each experiment was repeated in the figure legends. Because some of repeated data look better than previous ones, we replaced them in the revised manuscript (Figures 1E, 3B, 3I, and 5B).

2) The “direct” interaction between the Far complex and Atg32This has been well addressed by using recombinant proteins, showing that Far8 interacts directly with Atg32. Also the model has been adapted accordingly. These experiments clarify that the interaction between Atg32 and Far8 is direct. It seems however, that the interaction is way substoichiometric, as the interaction cannot be seen by coomassie staining for Far8? Why is this interaction so weak? How do the authors envision this works? This point should be addressed at least in the Discussion.

We thank the reviewer for this comment. Although the interaction between Atg32 and Far8 was slightly observed even by CBB staining for Far8, we did not include it in the original manuscript because the western blot data was much clearer than the CBB staining. For more accurate information, we included the CBB staining for Far8 in the revised manuscript (new Figure 6B).

As the reason for the weak interaction, we speculate; (i) Far8 preferentially interacts with phosphorylated Atg32 in vivo (Figure 5E), but we used non-phosphorylated Atg32 (produced in *E. coli*) for the GST pull-down assay.; (ii) The presence of Far3, Far7, and Far9 is likely to increase the interaction between Atg32 and Far8 (Figure 5F), but our in vitro assay used Far8 alone among six Far proteins.; (iii) Mitochondrial localization of Atg32 might be important for the stable interaction with Far8. We mentioned these points in the Discussion section.